# Heteroduplex oligonucleotide technology boosts oligonucleotide splice switching activity of morpholino oligomers in a Duchenne muscular dystrophy mouse model

Juri Hasegawa [1,2,9], Tetsuya Nagata [1,2,3,9] ✉, Kensuke Ihara [4,5], Jun Tanihata[6], Satoe Ebihara[1,2], Kie Yoshida-Tanaka[1,2], Mitsugu Yanagidaira[1,2], Masahiro Ohara [1,2], Asuka Sasaki[1,2], Miyu Nakayama [7], Syunsuke Yamamoto [7], Takashi Ishii[1,2], Rintaro Iwata-Hara [1,2], Mitsuru Naito [8], Kanjiro Miyata [8], Fumika Sakaue[1,2] & Takanori Yokota [1,2,3] ✉

The approval of splice-switching oligonucleotides with phosphorodiamidate morpholino oligomers (PMOs) for treating Duchenne muscular dystrophy (DMD) has advanced the field of oligonucleotide therapy. Despite this progress, PMOs encounter challenges such as poor tissue uptake, particularly in the heart, diaphragm, and central nervous system (CNS), thereby affecting patient's prognosis and quality of life. To address these limitations, we have developed a PMOs-based heteroduplex oligonucleotide (HDO) technology. This innovation involves a lipid-ligand-conjugated complementary strand hybridized with PMOs, significantly enhancing delivery to key tissues in *mdx* mice, normalizing motor functions, muscle pathology, and serum creatine kinase by restoring internal deleted dystrophin expression. Additionally, PMOs-based HDOs normalized cardiac and CNS abnormalities without adverse effects. Our technology increases serum albumin binding to PMOs and improves blood retention and cellular uptake. Here we show that PMOs-based HDOs address the limitations in oligonucleotide therapy for DMD and offer a promising approach for diseases amenable to exon-skipping therapy.

Duchenne muscular dystrophy (DMD), a lethal disease, is the most common form of myopathy, affecting ~1 in 3500 to 6000 live male births[1,2]. In patients with DMD, the lack of the dystrophin protein results in progressively severe muscle atrophy, consequent loss of ambulation, respiratory deficiency, cardiomyopathy, and premature death[3,4]. Dystrophin is an essential component of dystrophin-associated protein complex (DAPC) in the sarcolemma that has a key signaling role via neuronal nitric oxide synthase (nNOS) and other molecular components[5]. In the absence of functional dystrophin, DAPC components do not anchor to the sarcolemma. Exon skipping is a promising treatment for DMD, which enables the splicing mechanism

to bypass exon(s) to restore the translational reading frame and produce a internal deleted yet functional protein for out-of-frame mutations[6]. The recent FDA and PMDA approval of four phosphorodiamidate morpholino oligomer (PMO) therapies for the treatment of DMD has advanced the exon-skipping field[7–12]. The safety profile of PMOs has been investigated; they can be administered at high doses (80 mg/kg)[9,13]. Although treatment with PMOs has received considerable attention, their disease-modifying effect remains limited. Since the restored levels of the internal deleted dystrophin protein in skeletal muscles is 0.9–6% of the normal dystrophin level[9] and demonstrates poor tissue distribution, especially to the heart, DMD

constitutes an area of high unmet medical need[3,4]. Delivery and efficacy to the heart and diaphragm muscles, which have been difficult to achieve by naked PMO in preclinical studies, are particularly problematic[14–17]. Some patients with DMD show symptoms of developmental, cognitive, learning, and behavioral difficulties owing to the defect of dystrophin expressed in CNS[18], however since PMO cannot cross the blood-brain barrier (BBB), it is not expected to have a therapeutic effect on these symptoms.

Various challenges remain in the optimization of treatment regimens that must be overcome to provide clinical benefit for patients[3,4]. Therefore, the development of exon-skipping oligonucleotides with various backbone modifications, such as phosphoryl guanidine (PN)[19], tricyclo-DNA (tcDNA)[17,20], and/or different types of ligand conjugates containing transferrin receptor binder- conjugated antisense oligonucleotides (ASOs)[21,22] or peptide conjugate PMOs, remains essential[23–25].

Herein, we develop an exon-skipping therapy approach using the DNA/RNA heteroduplex oligonucleotide (HDO) technology to address existing limitations. Previously, we have reported a HDO technology[26] whereby HDO comprises a ASO duplexed with a complementary RNA strand conjugated to a lipid ligand for delivery. Compared with conventional ASOs, HDOs exhibit markedly improved blood retention and delivery to target organs and have achieved highly efficient gene silencing with different intracellular mechanisms[27,28].

In this study, we have developed a new type of HDO with a PMO in place of the gapmer-type ASO and a different intracellular mechanism from that of conventional HDOs, as RNase H cannot recognize PMO/HDO. We also assessed whether PMO/HDO improves treatment efficacy using a dystrophic *mdx* mouse model.

## Results
We initially designed PMO/HDO comprising PMOs targeting *Dmd* exon 23 and its complementary strand conjugated with alpha-tocopherol or cholesterol ligand on the 5′-terminal end (Fig. 1a), and confirmed the double-strand formation of both strands (Fig. 1b and Supplementary Table 1). The complementary strand of the PMO/HDO was not cleaved by RNase H, which differed from gapmer-type HDO[22], however, it was digested by RNase A (Supplementary Fig. 1a). PMO/HDO was readily dissolved in aqueous solutions. To quantify particle sizes, dynamic light scattering measurements of PMO and HDOs in PBS were conducted. The histograms for PMO and HDOs show a narrow size distribution with diameters of <10 nm (Supplementary Fig. 2b), suggesting that no aggregates have been formed. To investigate the pharmacokinetics (PK) of PMO/HDO in the serum of *mdx* mice, we administered systemic intravenous (IV) injections of single 100 mg/kg PMO doses or the molar equivalent of tocopherol-conjugated cRNA (11.88 μmol/kg) hybridized with PMOs (Toc-HDO) or cholesterol-conjugated cRNA (11.88 μmol/kg) hybridized with PMOs (Chol-HDO). The blood samples were collected at the indicated time points after a single injection and assessed using hybridization-based ELISA (HELISA)[29]. In this HELISA method, double-stranded PMO/HDO cannot be measured due to the presence of the complementary strand. Therefore, to measure all PMOs in the serum from *mdx* mice administered Toc-HDO or Chol-HDO, the collected samples were pretreated with RNase A to completely remove the complementary strand and convert them into naked PMOs before measuring them using the HELISA method. Although serum levels of the PMOs rapidly declined as previously reported (Fig. 1c)[29], the total PMO concentration of the RNase A treated serum from *mdx* mice administered Chol-PMO/HDO and Toc-PMO/HDO showed increased plasma retention (slower absorbance) and slower clearance compared with PMOs (Fig. 1c), similar to the gapmer-type HDO[27]. Compared to PMOs (AUC$_{0-24h}$, 3012.4 nM/h), Chol-HDO (AUC$_{0-24h}$, 21217.8 nM/h) exhibited a 7-fold greater drug exposure, while Toc-HDO showed a 5-fold greater exposure (AUC$_{0-24h}$, 15488.0 nM/h). Furthermore, the blood profile of

RNase A-untreated PMO/HDO indicates a slow dissociation of PMO from PMO/HDO in the blood (Supplementary Fig. 1C), rather than an immediate dissociation following injection.

To estimate the biodistribution of Chol-HDO and Toc-HDO to target tissues, we injected PMO at 100 mg/kg or the molar equivalent of Chol-HDO or Toc-HDO (11.88 μmol/kg) as the PK study into *mdx* mice and analyzed their content 48 h post-injection. PMO levels of Chol-HDO were 150- 280-fold higher than those in all tissues studied (Fig. 1d). To determine the mechanism underlying the enhanced PK in the blood and tissues, we evaluated the mouse albumin binding profile of PMOs, tocopherol, or cholesterol-conjugated PMO/HDO using a fluorescence polarization assay (Fig. 1e). Notably, HDOs showed highly significant enhancements in the binding affinity for albumin for which the parent PMO showed no affinity.

### Distribution of PMO via in situ hybridization
To evaluate the kinetic distribution of PMO or Chol-HDO in the heart and quadriceps femoris (QF) after systemic delivery, in situ hybridization was performed using a probe at 6 and 24 h, or 2 weeks following a single injection (11.88 μmol/kg). While a weak red signal (PMO) was detected in the heart and QF at 6 h post-administration of PMO alone, a strong PMO signal was detected in both tissues treated with Chol-HDO, consistent with the PK study results (Supplementary Fig. 1d, e). The localization of PMO from Chol-HDO-treated mice was not uniform across muscle fibers and was highly unevenly distributed in the QF. PMO from Chol-HDO-treated mice was distributed in normal-sized muscle fibers but was highly abundant, particularly in necrotic fibers and small-diameter fibers that appeared to be regenerating. At 24 h after a single injection, the PMO signals in both tissues treated with Chol-HDO rapidly decreased, especially in the heart. At 2 weeks post-injection, negligible PMO signal was observed in the heart and QF. Furthermore, we also conducted in situ hybridization chain reaction (ISH HCR)[30,31] to evaluate the localization of PMO (Cy5 probe) in the heart and QF 6 h after a single injection of either PMO or Chol-HDO at a concentration of 11.88 μmol/kg. To assess the nuclear localization, we simultaneously performed staining using antibodies against lamin A/C (labeled with Alexa 488) for the nuclear membrane and wheat germ agglutinin lectin (labeled with Alexa 594) for the cell membrane, along with DAPI counterstaining. In the heart and QF muscles treated with PMO, the PMO signal was scarcely detectable. In contrast, in the both muscles treated with Chol-HDO, there was a pronounced localization of PMO within both the cytoplasm and nuclei (supplementary Fig. 1f, g).

### PMO/HDOs induce dose–response and dose-frequency-dependent exon 23 skipping
We first performed a dose–response study of PMO and Chol-HDO. Mice were single intravenously injected with 10, 25, 50, or 100 mg/kg of PMO and the same equivalent molar of Chol-HDO (PMO) and sacrificed 2 weeks post-injection (Supplementary Fig. 2a). The heart, diaphragm, QF, and triceps brachii (TB) were harvested and analyzed for exon 23 skipping. On maximum, Chol-HDO treatment induced 6.1- 7.5-fold higher levels of skipping in skeletal muscles and the diaphragm and 22.7-fold higher levels in the heart than PMO. Dose-dependent curves of Chol-HDO showed that the skipping efficacy for the heart and diaphragm tended to increase with increasing doses. In skeletal muscles, meanwhile, the skipping efficacy of Chol-HDO tended to slightly saturate at ≥50 mg/kg doses. We also administered unliganded PMO/HDO at a dose of 100 mg/kg (11.88 μmol/kg). Similar to PMO, in the heart, there was almost no skipping activity, and in skeletal muscle and diaphragm, the skipping activity was slightly lower compared to PMO (Supplementary Fig. 2a red square). PMO conjugated directly with cholesterol could be synthesized. However, due to their high lipophilicity, they did not dissolve in water, thereby preventing their administration to mdx mice. This experience highlighted the benefit of

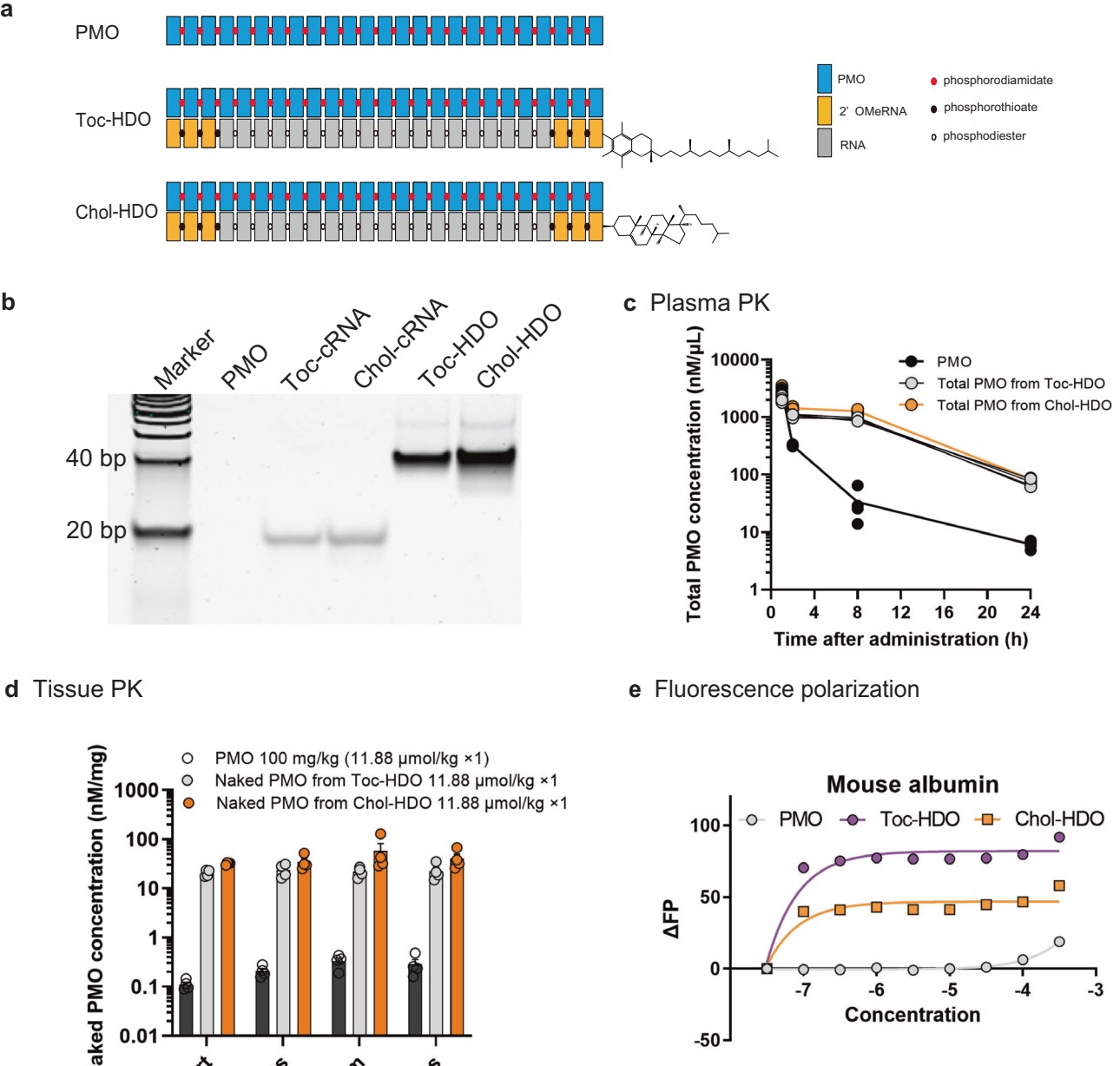

**Fig. 1 | Properties and biodistribution of PMOs and PMO/HDOs. a** Structure of phosphorodiamidate morpholino oligomers (PMOs) duplexed with lipid ligand (tocopherol (Toc) or cholesterol (Cho))-conjugated complementary strand. **b** Confirmation of annealing between PMOs and the complementary strand with lipid ligands electrophoresed on a 16% acrylamide gel. bp: base pairs (**c**, **d**) Pharmacokinetics of PMO after intravenous injection of a single 100 mg/kg (11.88 μmol/kg) PMO dose or molar equivalent of Toc-HDO or Chol-HDO. The hybridization-based ELISA shows the pharmacokinetic (**c**), and biodistribution (**d**) data in *mdx* mice (*n* = 4) injected with PMO, Toc-HDO or Chol-HDO. Data are presented as mean ± S.E.M. **e** Binding curves of PMO, lipid conjugated PMO/HDO with mouse albumin. Source data are provided as a Source Data file.

attaching lipids to PMO with a complementary strand, as in heteroduplex oligonucleotides.

To determine whether the skipping activity observed with the PMO/HDOs (Chol-HDO and Toc-HDO) and PMO could be further improved by multiple administrations, we intravenously injected mice once weekly for a total of 3 or 5 doses with 11.88 μmol/kg PMO or PMO/HDOs—the molar equivalent of 100 mg/kg of PMO—(Fig. 2a) and compared this with the single administration results. When PMO was administered at 100 mg/kg and injected once weekly for a total of 5 doses, approximately 2% exon-skipping was observed in the heart and

15–20% in the diaphragm and QF, paraspinal muscle, TB, and tibialis anterior (TA) (Fig. 2b). In contrast, mice treated with Toc-HDO using the same protocol showed ~33% skipping in the heart and 40% in the diaphragm and skeletal muscles (Fig. 2b).

Furthermore, consistent with PK analysis, in mice treated with Chol-HDO, approximately 45 % skipping was observed in the heart and 60–70 % in the diaphragm and skeletal muscles (Fig. 2b). On average, Chol-HDO treatment induced 4–5.5-fold higher levels of skipping in skeletal muscles and the diaphragm and 24.6-fold higher levels in the heart than PMO at the equimolar dosing regimen after five injections

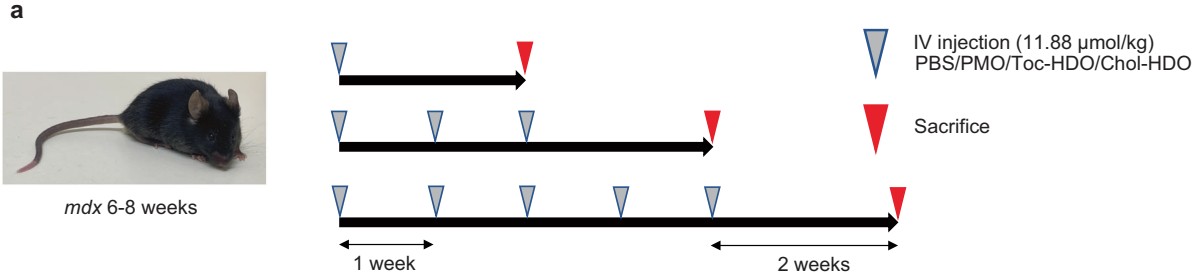

**a**

**b** exon 23-skipped dystrophin mRNA after 1, 3, or 5 x weekly administration

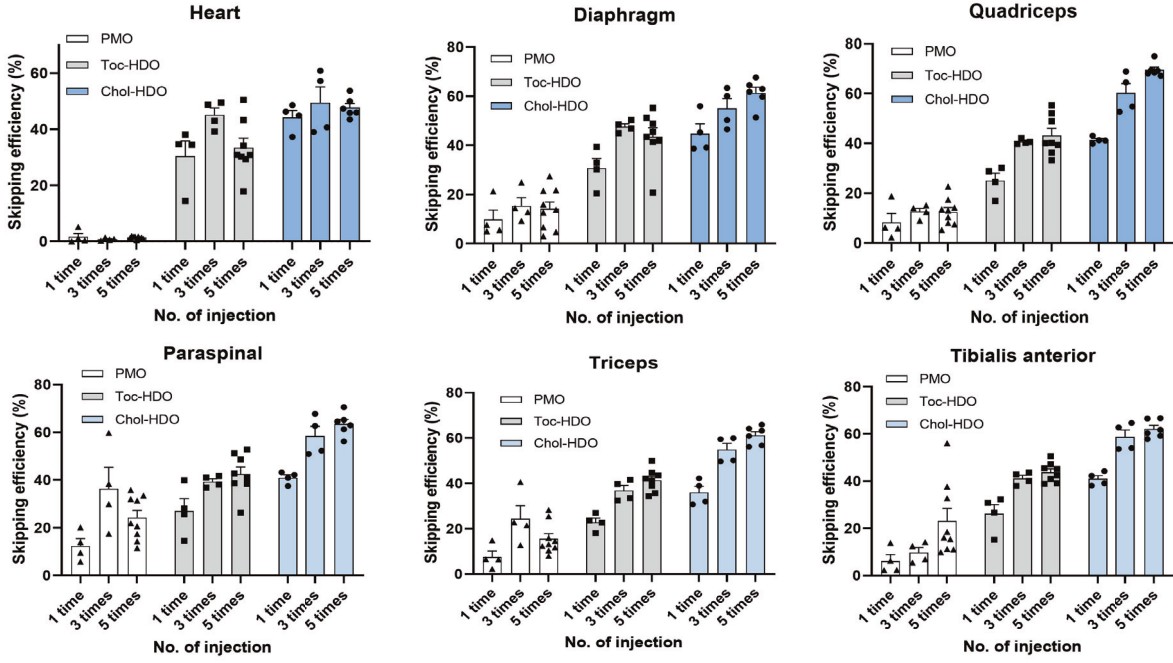

**c** Duration of exon 23-skipped dystrophin mRNA after 5 x weekly administration

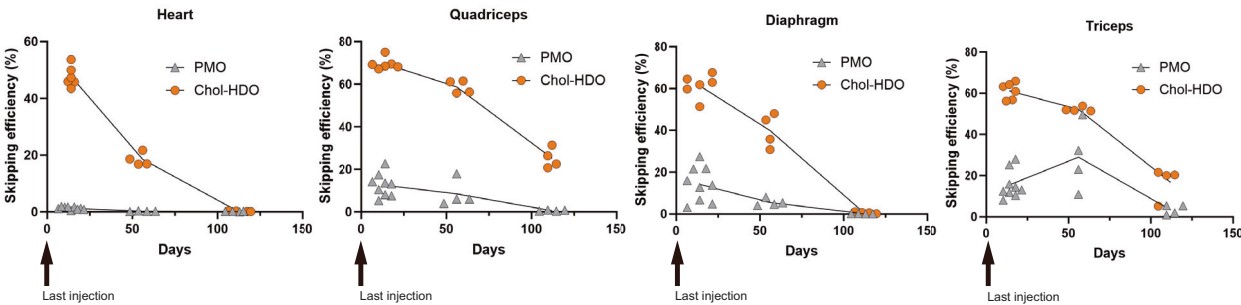

**Fig. 2 | Comparison of PMO or PMO/HDO effects on efficiency and duration of exon 23 skipping. a** Timeline of PMO or HDO administration and animal sacrifice. **b** Detection of exon 23-skipped dystrophin mRNA in the heart, diaphragm, and skeletal muscles of *mdx* mice 2 weeks after once-weekly systemic intravenous (IV) injections for a total of 1, 3, or 5 doses of PMO (100 mg/kg), Toc-HDO, or Chol-HDO

at a dose equimolar to PMO (11.88 µmol/kg). (*n* = 4–9 per group) Data are presented as mean ± S.E.M. **c** Time course of exon 23-skipped dystrophin mRNA 14, 56, or 112 days after five injections of PMO or Chol-HDO (11.88 µmol/kg) in indicated muscles of *mdx* mouse (*n* = 4–9 per group). Source data are provided as a Source Data file.

(Fig. 2b). High exon-skipping was also observed in other skeletal muscles, liver, or kidneys treated with Toc-HDO and Chol-HDO (supplementary Fig. 2B, C).

### Duration of PMO-HDO-induced exon 23 skipping
Sustained skipping is required for potential clinical applicability to extend the injection interval in patients with DMD. Therefore, we

evaluated the duration of exon skipping achieved with the lipid-conjugated PMO/HDOs. At 2 months after the fifth injection, the skipping efficiency in QF and TB remained at 80–90 % of that at 2 weeks, and ~35% of the skipping efficiency at 2 weeks was observed after 4 months. By contrast, the skipping efficiency (40–60%) of the heart and diaphragm at 2 months after five injections was attenuated earlier than that in the skeletal muscles. No skipping was observed in

the heart or diaphragm at 4 months following the final injection (Fig. 2c).

## Dystrophin restoration and improvement of histology by HDOs

We estimated the number of muscle fibers with dystrophin expression 2 weeks after five weekly injections of 11.88 μmol/kg and observed ~90% of dystrophin-positive fibers by Chol-HDO treatment (Fig. 3a, b) and 40–60% of dystrophin-positive fibers by Toc-HDO in the heart, diaphragm, QF, and TB (Supplementary Fig. 3A).

Hematoxylin and eosin (HE) staining showed the polygonization of muscle fibers and increased endomysial connective tissue between muscle fibers seen in *mdx* mice; however, these abnormalities were improved with HDO treatment (Supplementary Fig. 3B). To evaluate the effect of robust dystrophin expression on histology, the QF was double stained with Caveolin3 and DAPI (Fig. 3e), and then the myofiber cross-sectional area (CSA) and centrally nucleated fibers (CNF) of QF were measured as a marker of muscle regeneration[32,33]. For quantification, multiple images were acquired, and at least 5,000 fibers were counted per animal. As expected, the number of smaller fibers (<800 μm²) was reduced in Toc- or Chol-HDO groups relative to PBS-treated *mdx* controls, especially Chol-HDO, to almost the same distribution in B10 mice (Fig. 3c and Supplementary Fig. 3c). By contrast, the percentage of CNFs in the Chol-HDO groups was significantly decreased compared with that in PBS-treated *mdx* mice. The decrease in the CNF was mild[34] (Fig. 3d). The number of smaller fibers was increased in the *mdx* and PMO-treated *mdx* mice. We, therefore, investigated whether successful recovery of DAPC components to the sarcolemma occurred after the restoration of dystrophin by HDO. Relocalization of α-sarcoglycan, β-dystroglycan, and nNOS−component proteins of DAPC−was detected following HDO treatment (Supplementary Fig. 3D–F)[35].

Production of dystrophin was confirmed using western blot analysis (Fig. 3f). We found that PMO/HDOs restored markedly higher levels of dystrophin than PMOs in the heart and QF, where levels reached ~50% and ~100%, respectively, compared with those in wild-type B10 control mice.

## Chol-HDO treatment normalizes creatine kinase (CK) and motor function

Effective exon skipping and protein restoration of dystrophin at the molecular level led to phenotypic recovery in *mdx* mice. To assess muscle damage, we first estimated serum CK levels[32], a marker for muscle damage. CK was reduced to the normal level in mice treated with Chol-HDO after five IV injections and markedly reduced by Toc-HDO (Fig. 4a). This improvement in hyperCKemia by Chol-HDO persisted for 4 months following the final injection (Supplementary Fig. 4A, B).

The physiological properties were also studied by measuring forelimb grip power[32,36] and via the treadmill test[37]. A marked improvement in grip strength per body weight and treadmill endurance was observed in the Chol-HDO- and Toc-HOD-treated groups, compared with those in the *mdx* and PMO-treated groups (Fig. 4b, c). Particularly, Chol-HDO treatment normalized both functions of *mdx* mice to the level of those in B10 mice.

## Chol-HDO prevents QTc prolongation, QRS abnormalities, and fibrosis of the heart

We next evaluated the cardiac phenotype using an electrocardiogram (ECG), which highlighted significant differences between *mdx* and B10 control mice at an early stage. Young *mdx* mice display characteristic ECG changes observed in pre-clinical-stage human patients with DMD[38–40]; however, echocardiography only detects abnormalities after 40–43 weeks in *mdx* mice[41–43]. These ECG changes include a prolonged QRS duration, corrected QT interval (QTc) and QT interval, and tachycardia[38–40,44]. PBS- or PMO-injected *mdx* mice showed longer QRS

durations and QTc intervals than those in B10 mice at 12 weeks of age (Fig. 4d, e). These abnormalities in QRS durations and the QTc interval improved to normal levels in Chol-HDO-treated *mdx* mice 2 weeks after once weekly for a total of 5 doses of 11.88 μmol/kg, and the improvements persisted at 2 months following the final injection (Supplementary Fig. 4C).

Cardiac fibrosis is a characteristic trait of advanced cardiomyopathy in patients with DMD. The time at which fibrosis significantly increases in the heart of *mdx* mice compared with that of wild-type B10 mice reportedly varies from 4 months to 11 months[45–48]. When comparing B10 mice with *mdx* mice 2 months after once weekly for a total of 5 doses of PBS injections (4.5 months of age), *mdx* mice showed marked fibrosis in the cardiac tissue, similar to previously reported levels (Fig. 4f, g). Chol-HDO treatment in *mdx* mice ameliorated the fibrotic changes in the heart to the same level as in the B10 mice (Fig. 4f, g).

## Chol-HDO improves the restraint-induced fear response

Considering that we showed that Chol-HDO with the gapmer-type ASO penetrated the BBB with an efficient gene knockdown effect in CNS[27], the exon-skipping effect of PMO HDO in the whole brain was also examined. Similar to the level of exon skipping observed by Goyenvalle et al. using tricycloDNA was detected in the brain (Fig. 5a)[17]. We then measured the duration of the tonic immobility (freezing) and total movement distance traveled, both of which resulted from a restraint-induced fear response, a phenomenon reproducibly observed in *mdx* mice[17,49]. During the 10 min testing period, *mdx* mice treated with PBS or PMO showed ~80% of time freezing in response to this acute stress in contrast to only 50% for B10 mice (Fig. 5b). Chol-HDO treatment completely reverted the freezing time and movement distance to normal levels of the B10 mouse (Fig. 5b–d).

## Effect by subcutaneous administration of Chol-HDO

We then evaluated a subcutaneous (SC) route of administration. Five weekly SC injections of 11.88 μmol/kg Chol-HDO achieved 50–60% exon-skipping in the skeletal muscles and diaphragm and 20% in the hearts of those with IV injections (Fig. 6a). Significant improvements were also observed in the grip test and in serum CK levels, but not in treadmill test performance (Fig. 6b–d).

## Effect of the chemistry of complementary strands of Chol-HDO

To evaluate the significance of cleavability or stability of the complementary strands of Chol-HDOs, the effect of replacing the RNA in the center portion of the complementary strand with DNA or 2'-O-methyl (2'-OMe) RNA was examined (Fig. 7a and supplementary Table 1). The DNA in the center portion of the complementary strand showed a high skipping efficiency with marked improvements in CK levels and grip strength, similar to those with complementary RNA (Fig. 7b–d), suggesting that the center portion in the complementary strand can be cleaved by endogenous DNase as well as endogenous RNase in the heart and skeletal muscle cells. By contrast, full 2'-OMe modifications to the complementary strand lost their skipping activity (Fig. 7b–d). Since full 2'-OMe modifications of the complementary strand were stable >3 days post-intravenous injection in hepatocytes in vivo[26], intracellular cleavage of the complementary strand must be required to produce exon skipping in the heart and skeletal muscles.

## Adverse effects of multiple administration of HDOs

Both aspartate aminotransferase (AST) and alanine aminotransferase (ALT), the levels of which increase as a result of liver and muscle damage−as in the case of DMD and *mdx*[50,51]−were markedly elevated in the non-treated *mdx* mice (Supplementary Fig. 5A, B). A significant improvement was observed in serum AST/ALT levels in the mice after both Toc- and Chol-HDO treatments. Particularly in Chol-HDO, it was almost normalized, similar to that observed for CK levels

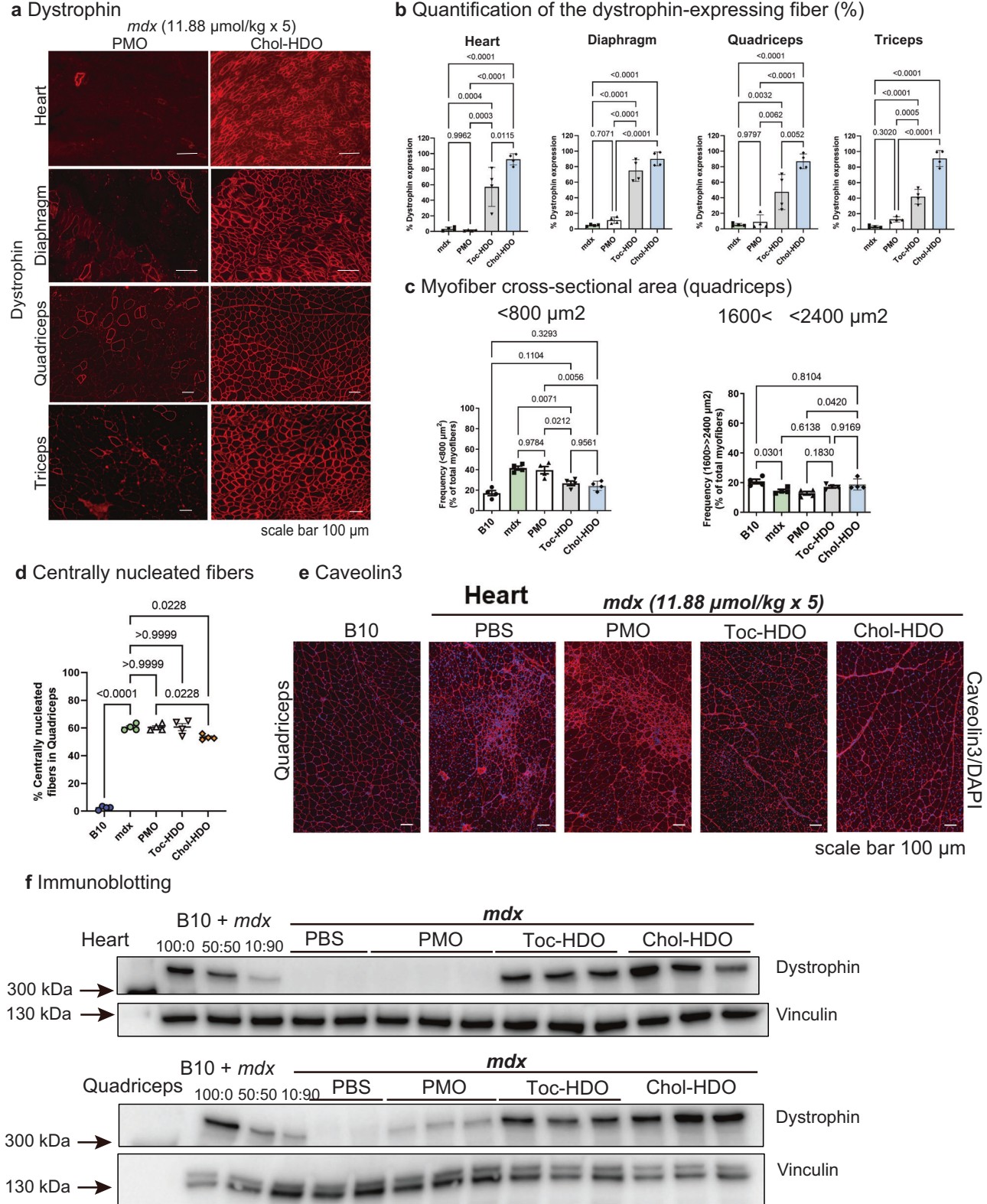

**Fig. 3 | Comparison of PMO or PMO/HDO effects on dystrophin expression.**
**a** Images of dystrophin immunostaining (red) in indicated muscles 2 weeks after the fifth PMO dose (100 mg/kg) or Chol-HDO at an equimolar dose to PMO (11.88 µmol/kg). Scale bar, 100 µm. **b** Quantification of dystrophin-expressing fibers (%) to the total number of fibers in the indicated tissues (*n* = 4 per group). **c** Myofiber cross-sectional area (CSA), and (**d**) percentage of centrally nucleated fibers (CNF) in quadriceps 2 weeks after five weekly injections of phosphate-

buffered saline (PBS), PMOs, or HDOs (11.88 µmol/kg) (*n* = 4 per group). **e** Representative images of caveolin 3 immunostaining (red) in quadriceps femoris (QF) counterstained with DAPI (blue) to evaluate CNF and CSA. **f** Western blot showing robust dystrophin expression in the heart and QF from HDO-treated mice. Data are presented as mean ± S.E.M and were analyzed using one-way analysis of variance followed by Tukey's tests (**b**–**d**). *P*-values are indicated. Source data are provided as a Source Data file.

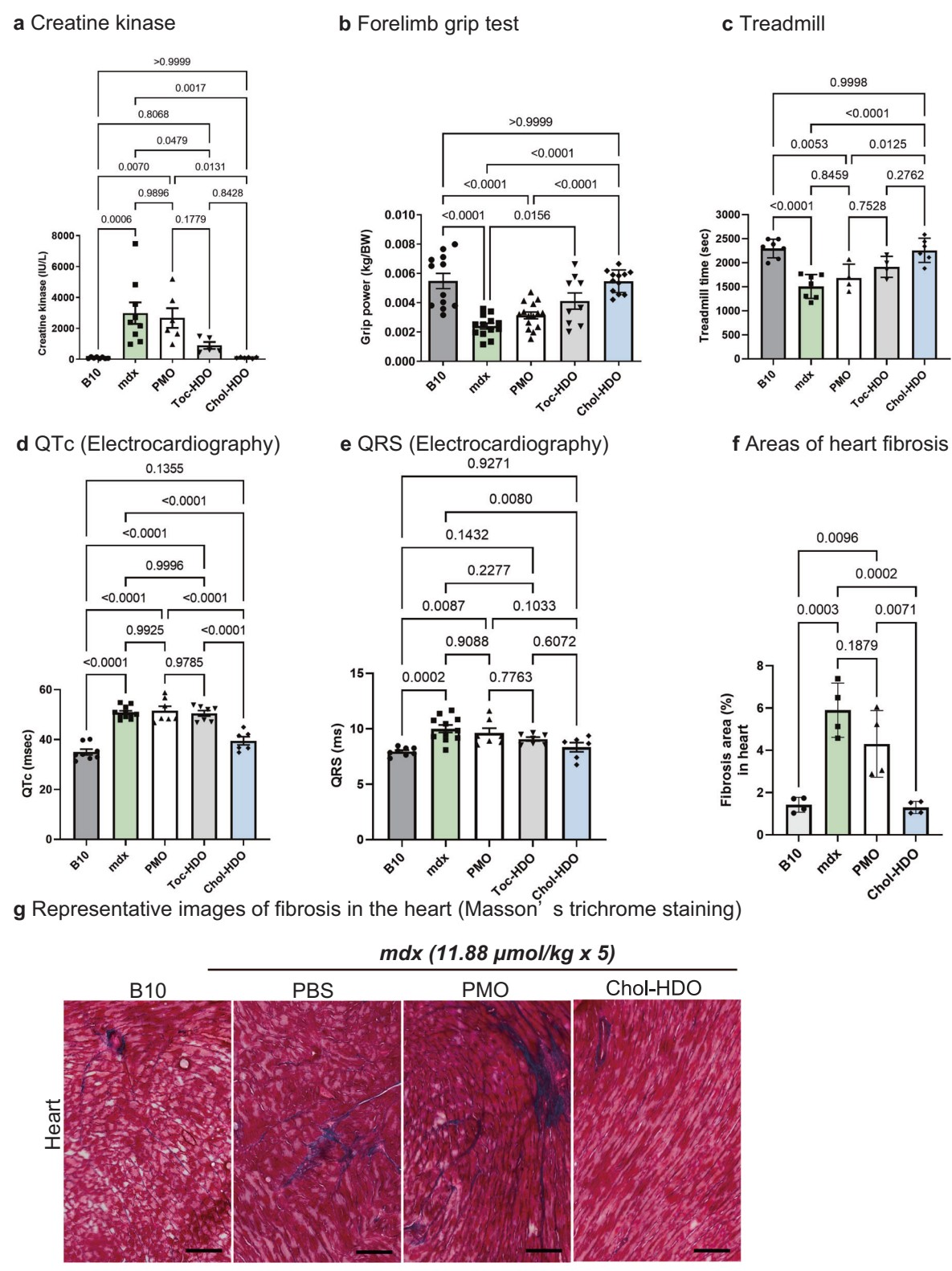

**Fig. 4 | Systemic delivery of PMO/HDOs drastically improves the *mdx* phenotype. a** Serum Creatine Kinase (CK) levels in mice injected once weekly for a total of 5 doses with PBS or HDOs (11.88 μmol/kg). Serum CK levels are reduced after treatment with Toc- or Chol-HDO, correlating with the levels of dystrophin restoration (*n* = 5-9 per group). **b** Forelimb grip test (*n* = 9–15 per group) and (**c**) treadmill test (*n* = 4–7 per group) performances evaluated in *mdx* mice injected once weekly for a total of 5 doses with PBS or HDOs (11.88 μmol/kg). ECG abnormalities observed in *mdx* are prevented in the treated *mdx* mice (**d**) QTc and

(**e**) QRS duration (*n* = 6–11 per group). **f** Quantification of the heart fibrosis-stained regions in the left ventricle (*n* = 4 per group). Transverse sections revealing the level of the papillary muscles 8 weeks (2 months) after once weekly for a total of 5 doses of PBS or HDOs (11.88 μmol/kg). **g** Representative images of the heart after Masson's trichrome staining in the left ventricle. Data are presented as mean ± S.E.M and were analyzed using one-way analysis of variance followed by Tukey's Kramer tests (**a**–**f**). *P*-values are indicated. Source data are provided as a Source Data file.

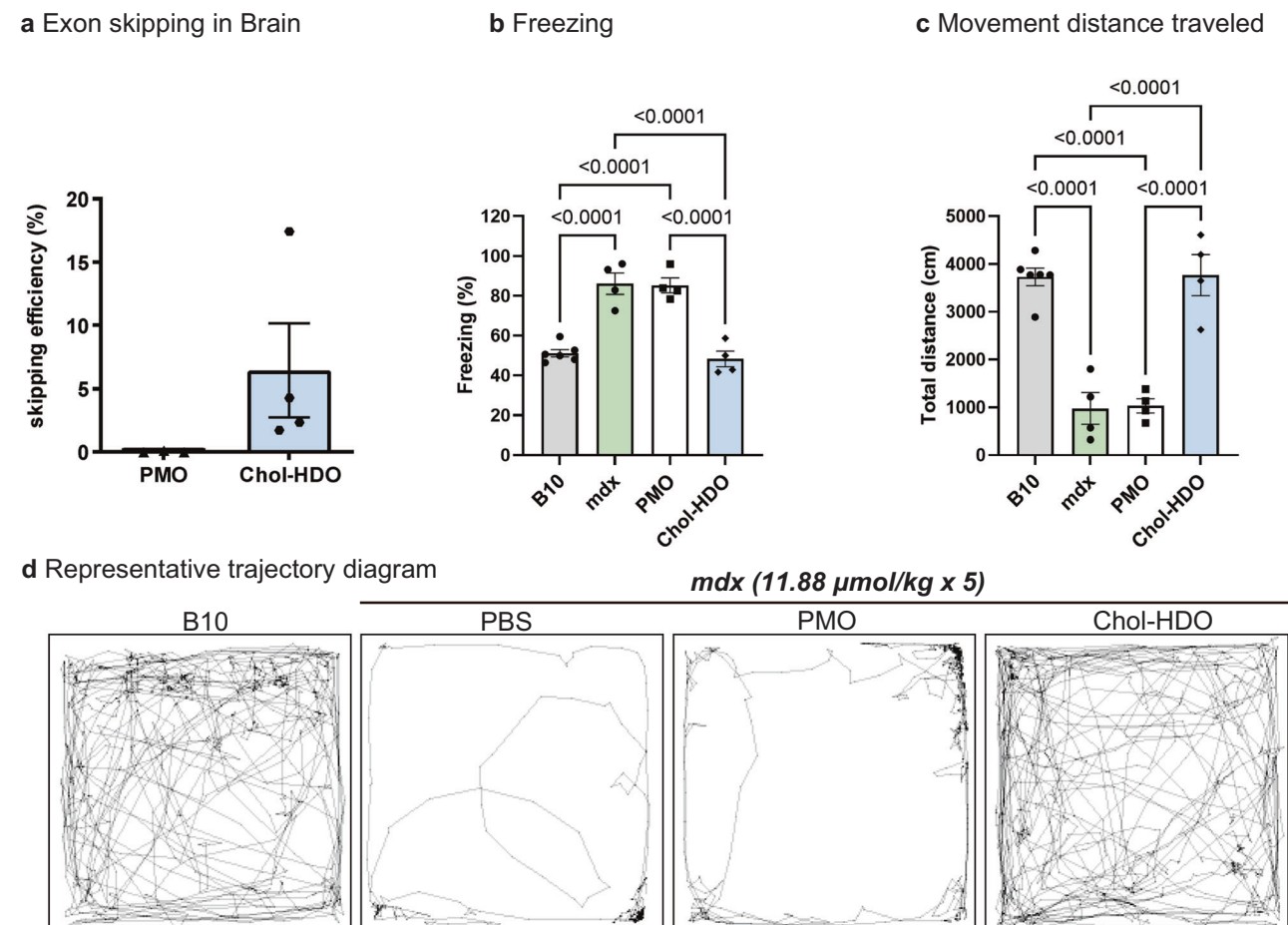

**a** Exon skipping in Brain    **b** Freezing    **c** Movement distance traveled

**d** Representative trajectory diagram

*mdx* (11.88 µmol/kg x 5)

B10    PBS    PMO    Chol-HDO

**Fig. 5 | Systemic delivery of PMO/HDOs drastically improves the restraint-induced fear response. a** Detection of exon 23-skipped dystrophin mRNA in the whole brain of *mdx* mice 2 weeks after once weekly for a total of 5 doses of PMO or Chol-HDO (11.88 µmol/kg) (*n* = 4 per group). **b** Duration of tonic immobility (freezing) expressed as a percentage of freezing time (*n* = 4–6 per group). **c** Total horizontal movement distance traveled (distance run in 10 min) (*n* = 4–6 per group). **d** Representative trajectory diagram of B10 and *mdx* mice treated with PBS, PMOs, or Chol-HDO. Data are presented as mean ± S.E.M. **a–c** And were analyzed using one-way analysis of variance followed by Tukey's Kramer tests (**b**, **c**). *P*-values are indicated. Source data are provided as a Source Data file.

(Supplementary Fig. 5A, B). In addition, renal function parameters, such as blood urea nitrogen (BUN) and serum creatinine (Cre) levels, were normal (Supplementary Fig. 5C, D). Highly efficient skipping activity was also observed in the liver and kidneys (Supplementary Fig. 2C) after multiple injections of Toc- or Chol-HDO, indicating that PMO/HDO was efficiently delivered to the liver and kidneys without any obvious toxicity (Supplementary Fig. 5A–D). Additionally, we have conducted histological evaluations using hematoxylin and eosin staining of *mdx* mice treated with PBS, PMO, Toc-HDO, or Chol-HDO (Supplementary Fig. 5E). In the liver pathology (upper panel), small foci of inflammatory cell infiltration were observed in the liver parenchyma of *mdx* mice treated with any of the interventions, including PBS. No other significant lesions were observed in *mdx* mice treated with Chol-HDO. However, in *mdx* mice treated with Toc-HDO, there was occasional increased size heterogeneity of hepatocyte nuclei, sometimes accompanied by meganucleation. In kidney pathology (lower panel), no significant lesions were noted in *mdx* mice following treatment with PBS or Chol-HDO. On the other hand, *mdx* mice treated with Toc-HDO occasionally showed a slight increase in cellular density in glomeruli.

## Discussion

Herein, we developed a new type of HDO technology using PMOs for the heart, diaphragm, skeletal muscle, and CNS. Both Chol- and Toc-HDO treatments resulted in marked improvements in motor function, cardiac phenotype, muscle pathology, and serum CK levels. In

particular, Chol-HDO normalized the motor and cardiac phenotype, serum CK, and muscle pathology in *mdx* mice to their levels in B10 mice, although the cardiac phenotype of *mdx* mice is very mild. Approximately 80–90% of muscle fibers in the heart, diaphragm, QC, and TB of the Chol-HDO-treated group expressed dystrophin, as evidenced by immunostaining; immunoblotting analysis revealed the expression of dystrophin in the heart and QC at >50–100% of the expression observed in B10 mice. There were no small diameter fibers that regenerated or necrotic fibers, neither was there a widening of the interstitial tissues. In addition, the robust restoration of dystrophin was accompanied by elevated expression of α-sarcoglycan, β-dystroglycan, and neurogenic nitric oxide synthase—components of the dystrophin-associated glycoprotein complex (DAPC)—in the sarcolemma. The restoration of sarcolemmal nNOS observed here may have an additional effect, as nNOS expression counteracts vasoconstriction and focal ischemia following exercise that further damages the DMD muscle[52,53]. Moreover, the normalization of cardiac dysfunction with robust expression of dystrophin in the heart of *mdx* mice indicate a potential for improved prognosis in patients with DMD.

Dystrophin is also expressed in CNS within specific regions of the brain[18]. Enhanced freezing behavior in response to restraint stress, which is not painful but induces psychological stress, was observed in *mdx* mice[49]. The present results show that Chol-HDO improved freezing and the movement distance traveled in response to restraint in the *mdx* mice. This suggests that the expression of dystrophin

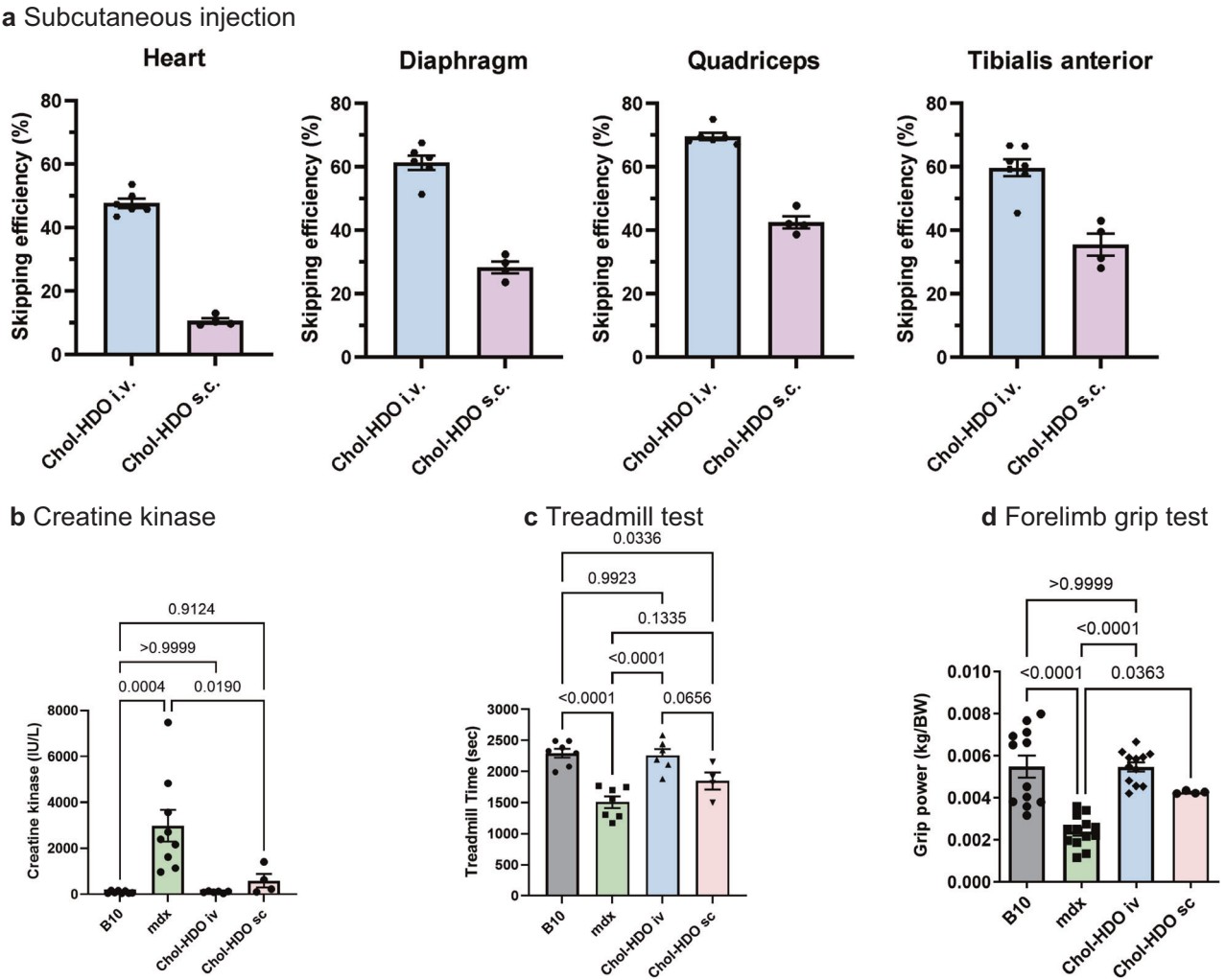

**Fig. 6 | Effects of subcutaneous administration of Chol-HDO and complementary strand modification on exon-skipping efficiency and functions.** **a** Detection of exon 23-skipped dystrophin mRNA in the indicated muscle of *mdx* mice after five weekly SC injections of Chol-HDO (11.88 μmol/kg) (*n* = 4–7 per group). **b** Serum CK levels in mice with five weekly SC injections of Chol-HDO (11.88 μmol/kg) (*n* = 4–9 per group). **c** Treadmill test (*n* = 4–7 per group) and (**d**)

forelimb grip test (*n* = 4–13 per group) were also evaluated in *mdx* mice subcutaneously injected five times with Chol-HDO (11.88 μmol/kg). Data are presented as mean ± S.E.M (**a**–**d**) and were analyzed using one-way analysis of variance followed by Tukey's Kramer tests (**b**–**d**). *P*-values are indicated. Source data are provided as a Source Data file.

contributes to this normalization not only in the skeletal muscle but also in the CNS, as Chol-HDO could cross the BBB. Improving the freezing behavior could be associated with improving CNS/psychological symptoms in patients with DMD.

Various mechanisms likely underly the effects elicited by lipid-conjugated PMO/HDO. First, the blood retention by PMO/HDO was markedly improved with an AUC 5–7 fold higher than that of the parent PMO as with the gapmer type-HDO[27]. The RNase A-treated HELISA results revealed that nearly 90% of PMO/HDO were in a double-stranded configuration in the blood 1–2 h after administration. Second, the delivery of PMOs to target muscles (Tissue PK) by PMO/HDO was markedly enhanced by 100–150-fold compared with the parent PMO. Moreover, the PMO signal in the muscle tissues of the PMO/HDO group, as observed following PMO in situ hybridization (ISH), was also markedly higher in the PMO/HDO group than in the PMO group, indicating an improved cellular uptake effect that cannot be explained by its increased blood retention. Additionally, the fluorescence polarization (FP) results indicate that the binding to albumin was markedly enhanced in PMO/HDO as compared to that in the parent PMO. The increased binding to albumin may have increased PMO/HDO blood retention by limiting glomerular filtration and urinary excretion.

Chappell et al. previously reported that increased affinity for serum albumin by lipid-conjugated ASO facilitates ASO transport across endothelial barriers into the interstitium of the muscle[54]. The increased blood retention of PMO/HDO also increased the exposure to endothelial cells, and an increased affinity for serum albumin may have facilitated transfer through the endothelial barrier into the muscle interstitium, resulting in improved muscle uptake. In addition to improved blood retention, the binding of blood lipoproteins by the cholesterol ligand may have enhanced their uptake into muscle cells[55]. Particle formation was suggested to be involved in the enhanced uptake of tcDNA[17] and peptide PMO[56] into the tissues; however, the DLS results showed that no particle formation was observed in PMO/HDO.

Cholesterol and tocopherol-conjugation increased PMO delivery to the liver and kidneys. However, no obvious toxic effects, such as altered serum hepatic or renal function indices, or adverse clinical outcomes were observed during the course of our experiments (up to 4 months after the last injection), even following administration of multiple high doses (100 mg/kg; 11.88 μmol/kg).

An improvement in serum CK levels was observed 2 and 4 months after the last injection, and an improvement in ECG was observed

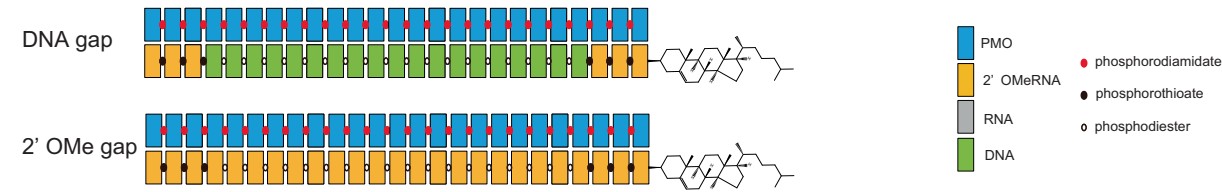

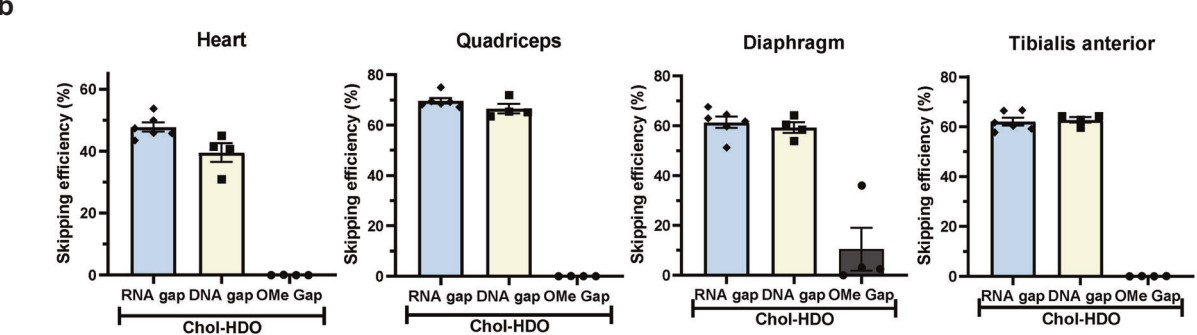

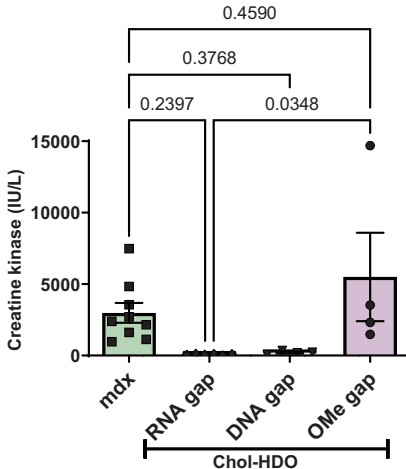

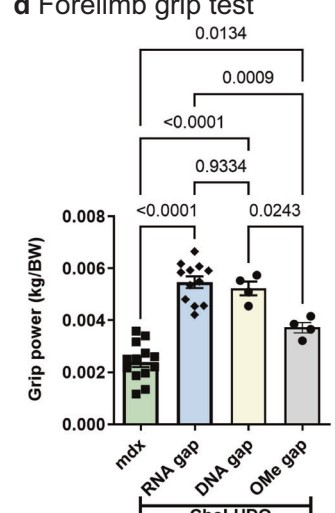

**Fig. 7 | Effects of complementary strand modification on exon-skipping efficiency and functions. a** Structure of the cholesterol-conjugated complementary strand DNA gap and 2'-OMe gap. **b** Detection of exon 23-skipped dystrophin mRNA in the indicated tissues of *mdx* mice 2 weeks after weekly injections for a total 5 doses of Chol-HDO (11.88 μmol/kg) with DNA gap or 2'-OMe gap (*n* = 4–6 per group). **c** Serum CK levels (*n* = 4–9 per group) and (**d**) forelimb grip test (*n* = 4–13 per group) results in mice treated with Chol-HDO (11.88 μmol/kg) with DNA gap or 2'-OMe gap. Data are presented as mean ± S.E.M. **b**–**d** and were analyzed using one-way analysis of variance followed by Tukey's Kramer tests (**c**, **d**). *P*-values are indicated. Source data are provided as a Source Data file.

2 months after the last injection. Although weekly dosing was used in this study for comparison with PMOs, the dosing interval could be further extended or the single dose could be reduced due to the robust and sustained expression of dystrophin as well as its long half-life[57,58]. Few studies have evaluated the effects of SC PMO administration in adult animals[59,60]. In the current study, SC administration of PMO/HDO induced a slightly weaker skipping effect than IV administration; however, improved functioning was observed with the former. Thus, long-term SC administration is expected to have higher efficacy and provides the option of self-administration. Collectively, these results indicate that HDO technology may represent a new avenue for novel exon-skipping drugs for DMD and other multisystemic disorders.

The basic concept of HDO originally assumed that the complementary strand is cleaved by RNase H in the cell. However, as shown in Supplementary Fig. 1A, the complementary strand of this new type of HDO might be cleaved by other RNases, such as RNase A and RNase T2, in the endosome or nucleus. DNase would cleave the complementary strand in case of use natural DNA instead of natural RNA in center portion of the complementary strand. Additionally, PMO/HDOs consisting of a complementary strand fully composed of 2'OMe modifications showed no in vivo skipping activity, likely because the complementary strand was not cleaved owing to the high resistance of 2'-OMe RNA to nucleases. In our results, however, the increased skipping efficiency achieved by single dosing may not correspond with the extreme increase in PMO concentration (100–150-fold) within the muscles of mice treated with PMO/HDO. We initially postulated that the complementary strand separation was poor. However, most of the complementary strand was likely already separated from the PMO/

HDO in the muscle tissue since ISH and HELISA use the complementary strand of the PMO sequence as a probe and binds only to single-stranded PMO. In view of this, we propose two possible explanations. First, the endosomal escape of PMO from PMO·HDO may be inefficient, resulting in low migration to the nucleus. Second, increased delivery into necrotic fibers might be unproductive. In the ISH data of the QF, PMO was distributed in normal-sized muscle fibers but was highly abundant, especially in necrotic fibers and small-diameter fibers that appeared to be regenerating fibers. However, after five injections of Chol-HDO, necrotic and regenerating fibers were relatively absent, suggesting that long-term administration may decrease the concentration of PMO in the QF, as it was taken up by normal-sized muscle fibers. Therefore, ligands must be developed that will be preferentially taken up by normal-sized muscle fibers.

In conclusion, we have developed a new type of lipid-conjugated HDO using parent PMOs, resulting in a functionally normal motor phenotype in a mouse model of DMD, including the normalization of abnormalities in cardiovascular and behavioral symptoms. Although further optimization of intracellular complementary strand cleavage is necessary, these PMO/HDO properties make it particularly attractive as a treatment for patients with DMD and other genetic diseases affecting the heart, CNS, and skeletal muscles, who are eligible for exon-skipping therapy.

## Methods

### Antisense oligonucleotides
PMOs were synthesized by Gene Tools, LLC (Philomath, OR). All complementary strands for the experiment were synthesized by GeneDesign (Osaka, Japan). PMOs target the donor splice site of exon 23 (+7–18) of the mouse dystrophin pre-mRNA, 5′-GGCCAAACCTCGGCTTACCTGAAAT-3′.

### Mouse studies
All animals were maintained on a 12 h light/12 h dark cycle in a pathogen-free animal facility (temperature: 18–24 °C; humidity: 40–70%) with free access to food (CLEA Rodent Diet CE-2, (CLEA Japan, Inc., Japan)) and water ad libitum. Mice (mdx, C57BL/10ScSn-Dmdmdx/J, 6–8 week-old males, and C57BL/10ScNJic [B10], 6–8 week-old males) were injected intravenously in the retro-orbital sinus or subcutaneously once per week with AONs, under general anesthesia using isoflurane. They were randomly assigned to experimental or control groups. All studies were conducted in accordance with the ethical guidelines of Tokyo Medical and Dental University, and in strict compliance with the Fundamental Guidelines for Proper Conduct of Animal Experiment and Related Activities in Academic Research Institutions as set forth by the Ministry of Education, Culture, Sports, Science and Technology. Approval for the experiments was granted by TMDU (Approval number A2022-085A). Experiments are in accordance with the ARRIVE guidelines. All possible efforts were made to minimize the number of animals used and to alleviate their discomfort.

### Estimation of skipping efficiency in vivo
Each antisense oligonucleotide (AON) against exon 23 of the dystrophin gene was dissolved in PBS (stock concentrations: 2 mM), and 11.88 μmol/kg of each AON was injected into the retro-orbital sinus once weekly for a total of 1, 3, or 5 doses. Two weeks after the last injection, mice were sacrificed under anesthesia with 4% isoflurane (Wako, Osaka, Japan), and the muscles, brain, liver, and kidneys were dissected. Total RNA was extracted from cells or muscle tissues using ISOGEN 2 (NIPPON GENE, Tokyo, Japan), and 300 ng or 500 ng of total RNA was processed using the QIAGEN OneStep RT-PCR Kit (QIAGEN, Venlo, Nederland), according to the manufacturer's instructions. The primer sequences were mEx22F 5′-ATCCAGCAGTCAGAAAGCAAA-3′ and mEx24R 5′-CAGCCATCCATTTCTGTAAGG-3′ for amplification from exons 22 to 24. The PCR conditions were 50 °C for 30 min and

95 °C for 15 min, 35 cycles of 94 °C for 1 min, 60 °C for 1 min, 72 °C for 1 min, and finally, 72 °C for 7 min. The PCR bands were analyzed using Bioanalyzer 2100 (Agilent, Santa Clara, CA, USA), and the resulting PCR bands were extracted using a QIAquick Gel extraction Kit (QIAGEN, Venlo, Nederland) for direct sequencing using an ABI 3100 (Thermo Fisher Scientific, Waltham, MA, USA) to confirm the exon skipping. Skipping efficiency was calculated using the following formula:

[(molality of skipped translation products) × 100% / (molality of skipped translation products + molality of unskipped translation products)].

### Immunohistochemistry
Ten-micrometer cryosections were cut from flash-frozen muscle using the Leica CM3050 S, (Leica, Wetzlar, Germany) placed on MAS-coated glass slides (Matsunami Glass Industrial, Osaka, Japan), air-dried, and blocked for 1 h with 5% goat serum (S-1000 Vector Laboratories, Burlingame, CA, USA) in PBS or mouse-on-mouse blocking buffer containing mouse IgG blocking reagent (#MKB-2213 Vector Laboratories) at room temperature (~25 °C). The tissues were then incubated with the following primary antibodies overnight at 4 °C: rabbit anti-dystrophin against C-terminus (ab15277, 1:300; Abcam), mouse anti-α-sarcoglycan (NCL-L-a-SARC, 1:200; Leica Biosystems), mouse anti-β-dystroglycan (NCL-b-DG, 1:200; Leica Biosystems), nNOS (#61-7000-rabbit, 1:1000; Thermo Fisher Scientific), and mouse anti-Caveolin 3 (sc-5310, 1:500; Santa Cruz Biotechnology). Subsequently, tissue sections were treated with secondary antibodies (Alexa Fluor 546 goat anti-mouse #A-11030 and Alexa Fluor 568 goat anti-rabbit #A-11011 Thermo Fisher Scientific) for 1 h (1:1000), protected from light at room temperature. Coverslips were mounted using VECTASHIELD Antifade Mounting Medium with 4′,6-diamidino-2-phenylindole (DAPI) (VECTOR H-1200). Centrally nucleated fibers and the myofiber cross-sectional area of QF were measured using HALO® Image Analysis (Indica labs, Albuquerque, NM, USA) (28).

### In situ hybridization to the morpholino oligomer
In situ hybridization of the morpholino oligomer was performed using the miRNAscope® HD (RED) Assay Kit (Advanced Cell Diagnostics [ACD], Westminster, CO, USA), according to the manufacturer's instructions. Fresh-frozen QF muscles and heart tissues were sectioned (10 μm) using the Leica CM3050 S, (Leica, Wetzlar, Germany) and placed on SuperFrost Plus slides (Thermo Fisher Scientific). Slides were fixed in 4% paraformaldehyde for 1 h at 4 °C, incubated in 50% ethanol for 5 min, 70% ethanol for 5 min, and washed in 100% ethanol twice for 5 min each. Sections were then incubated in hydrogen peroxide for 10 min at room temperature and washed in distilled water twice for 1 min each. Protease IV treatment was applied to the tissues, which were incubated in a chamber at room temperature for 30 min, followed by washing with PBS for 1 min. Slides were incubated with PMO sequence probes (SR-ASO-PMO-S1, #1088271-S1, ACD) for 2 h at 40 °C. Further amplification of the target probe signal was performed according to the manufacturer's instructions (miRNAscope HD detection protocol Amp 1-6). Fast red was prepared by combining Red-A and Red-B (1:60), added to the sections, and incubated for 10 min at room temperature. Slides were mounted with EcoMount (EM897L, Biocare, Pacheco, CA, USA), counterstained with hematoxylin, and imaged on SLIDEVIEW VS200 (Evident Co., Tokyo, Japan) at ×40 magnification.

### In situ hybridization chain reaction (HCR)
Tissue sections were prepared from formalin-fixed paraffin-embedded blocks. Deparaffinization of slides was performed in xylene and ethanol solutions at room temperature (25 °C). The sections were pre-treated in Tris-EDTA Buffer, pH 9.0 (ab93684, abcam, Cambridge, UK) with microwave oven for 5 min and another 5 min for antigen retrieval. The sections were left immersed in the buffer at room temperature for

at least 20 min and then slowly cooled. The slides were block with 3% bovine serum albumin (017-22231, FUJIFILM Wako Pure Chemical Corporation, Osaka, Japan) in phosphate-buffered saline for 15 min. The mixture of anti-lamin A/C antibody (1:100, 8617 s, Cell Signal Technology, MA, US) and wheat germ agglutinin lectin (1:300, W11262, Thermo Fisher Scientific Inc., MA, US) was added to the slides and incubated overnight at 4 °C to stain nuclear membrane and cell boundary. In situ hybridization chain reaction was conducted to evaluate the distribution of PMO as Zhuang et al. with minor modifications[30]. Briefly, the hybridization of probe to PMO was performed according to the miRCURY® LNA® miRNA ISH Optimazation Kits (FFPE) protocol (Qiagen, Hilden, Germany). The LNA-modified probe with overhang for signal amplification was designed and synthesized at Qiagen. The sequence is 5'-CTCTATATCT CCAACCCGAATTTCAGGTAAGCCGAGGTTT-3'. Slides were washed in 5X SSCT for 10 min and then incubated in amplification buffer (5X SSCT, 0.1 % Tween 20, 10% low molecular weight dextran sulfate) for 30 min at room temperature Hybridization chain reaction was performed in amplification buffer containing 6 µmol/L hairpin amplifiers. Slides were mounted in Prolong Diamond with DAPI (P36966, Thermo Fisher Scientific Inc.). Sections were imaged on an STELLARIS 8 confocal microscope (Leica Microsystems, Wetzlar, Germany). Imaging analysis was conducted with Imaris (ver. 9.7.0, Oxford Instruments, Abingdon, UK).

## Western blotting

Proteins were extracted from sliced frozen muscle using SDS buffer (0.125 M Tris/HCl with pH 6.4, 10% glycerol, 4% SDS, 4 M urea, 10% ß-ME, and 0.005% BPB) supplemented with 1X Protease Inhibitor (Complete Mini, Roche Diagnostics, Mannheim, Germany). The normal control lysate from a B10 mouse was prepared as a reference for dystrophin expression. Subject and normal control lysates were denatured at 100 °C for 3 min and electrophoresed in a Tris-acetate 3−8% gradient polyacrylamide gel (Thermo Fisher Scientific) at 150 V for 40 min. The proteins were transferred to a PVDF membrane (Bio-Rad, Hercules, CA) via wet transfer at 30 V overnight. After incubation with 5% nonfat milk (NACALAI TESQUE, INC., Kyoto, Japan) in TBST for 60 min, the membrane was incubated at 4 °C overnight with an anti-dystrophin antibody (ab15277, 1:200; Abcam, Cambridge, UK) or anti-vinculin antibody (NB600-1293, 1:10000; Novus Biologicals, Centennial, CO, USA). The membrane was washed three times for 10 min each in TBST and incubated with a horseradish peroxidase-conjugated anti-rabbit (#111-035-003, 1:3000) or anti-mouse (#115-035-003, 1:10,000) antibodies (Jackson ImmunoResearch, West Grove, PA, USA) for 60 min, followed by six washes with TBST and allowed to develop with West Dura Extended Duration Substrate (Thermo Fisher Scientific), according to the manufacturer's protocols. The immunoreactive bands were detected using the ChemiDoc XRS Image System (Bio-Rad Laboratories, Hercules, CA, USA).

## Measurement of the PMO concentration

HELISA was performed based on the method previously reported by Burki et al.[29] and Lim et al.[61]. PMOs in the blood were quantified using sera from blood samples of treated *mdx* mice or age-matched samples. For PMO uptake quantification, frozen muscle sections were weighed, homogenized in RIPA buffer (Thermo Fisher Scientific), and incubated with proteinase K (NACALAI TESQUE, INC., Kyoto, Japan) overnight at 55 °C. Then, lysates were spun at maximum speed for 15 min to collect the supernatant. Probes with complementary sequences to the PMOs used were synthesized and conjugated at the 5' and 3' ends with digoxigenin and biotin, respectively (GeneDesign, Osaka, Japan). The first and last seven nucleotides of the probes were fully phosphorothioated. PMO amounts were calculated in reference to a standard curve constructed from fluorescence values given by the respective PMO standards.

## Estimation of fibrosis

Frozen tissue sample sections were stained with the Trichrome Stain Kit (Modified Masson's: ScyTek Laboratories), according to the manufacturer's instructions. Sections were stained with preheated Bouin's Fluid for 60 min, cooled for 10 min, and washed twice with water. Then, slides were stained with working Weigert's Iron Hematoxylin for 5 min, washed with water for 2 min, and Biebrich Scarlet/Acid Fuchsin Solution was applied to slides for 15 min. After washing with water, slides were differentiated in Phosphomolybdic/Phosphotungstic Acid Solution for 10−15 min, followed by Aniline Blue Solution for 5−10 min, and washed in water. Images were obtained by SLIDEVIEW VS200 (Olympus, Japan) at × 20 magnification. Blue areas of the transverse *mdx* mouse heart sections were measured at the papillary muscle level as areas of fibrosis; these values were then divided by the total cross-sectional area to determine the degree of cardiac fibrosis using HALO® Image Analysis (Indica labs, Albuquerque, NM)[62].

## Muscle function analysis

**Forelimb grip test.** Muscle strength was measured using the forelimb grip test with a grip strength meter (MK-380CM/FM; Muromachi Kikai, Co., Ltd., Tokyo, Japan). The average of three measurements per animal per time point was recorded for comparative analysis.

## Treadmill exercise

Running sessions were performed on a four-lane motorized treadmill equipped with electric shock (Treadmill for Rats and Mice Model MK-680 S; Muromachi Kikai Co., Ltd) at least 1 week after the last injection. The treadmill was set at an inclination of 0°. All mice were acclimated to the treadmill belt for 5 min before starting to walk and then forced to run at 5 m/min for 5 min. Subsequently, the speed was increased by 1 m/min each minute. The test was stopped when the mouse was exhausted, did not attempt to remount the treadmill, or spent 5 continuous seconds on the shock grid, and the time to exhaustion was determined.

## Restraint-induced unconditioned fear

All mice were tested between 10:00 am and 1:00 pm. Mice were restrained by the experimenter by placing the neck between the thumb and index finger and positioning the tail between the third and little fingers. After 10 s, the mouse was released into the observation box illuminated with 20 lux. Videos were captured by a camera placed on top of the box for 10 min using Image OF (O'Hara & Co. Ltd., Tokyo, Japan) and then transmitted to a personal computer (Panasonic, Japan). Total distance traveled and immobile time were measured. Complete immobilization of the mouse, except for respiration, was regarded as a freezing response[63]. This was quantified as the time the mouse moved <0.5 cm (2 cm) per second. Unconditioned fear responses induced by this acute stress were characterized by periods of tonic immobility (freezing) during the 10 min recording period.

## Electrocardiography

Body-surface electrocardiography (ECG) was performed in a blinded manner, as described previously[63]. ECG in lead II configuration was recorded using the PowerLab system (PowerLab 4/26, ADInstruments) under anesthesia with 1% isoflurane. ECG parameters were obtained by averaging those from three different ECGs. The QT interval was defined as an interval between the onset of the QRS complex and the end of the negative component of the T wave. QTc was calculated using the following formula: QTc = QT interval (ms)/√(RR interval (s) × 10).

## Blood chemistry and complete blood count analysis

Blood chemistry was assessed in the SRL Laboratory (Tokyo, Japan), and the blood cell count was measured at LSI Medicine (Tokyo, Japan).

## Size and zeta-potential measurements

The size and size distribution of nanoparticles were determined via DLS using a Zetasizer Pro instrument (Malvern Instrument Ltd., UK) with an incident light (633 nm). The sample solutions were loaded into a low-volume cuvette (ZEN2112), and the measurements were carried out with a detection angle of 173° and a temperature of 25 °C.

## Fluorescence polarization

Binding of PMOs and lipid-conjugate PMOs/HDOs to mouse albumin was determined using fluorescence polarization (FP), as described by Gaus et al.[64]. Briefly, PMO and PMO/HDO were labeled at the 5′ terminus of the PMO with Alexa Fluor 647. Binding measurements were conducted in 1X DPBS (Gibco) in flat-bottom non-binding 96-well plates (Corning, NY, USA) at 25 °C. Alexa 647-labeled PMO or PMO/HDO were added at a final concentration of 2 nM to solutions of albumin ranging from sub nM to mM concentrations. Solutions were equilibrated at least 30 min before measuring fluorescence polarization ($\lambda ex = 635$ nm, $\lambda em = 675$ nm) on a Tecan InfiniteM1000 Pro (Baldwin Park, CA, USA).

## Statistical analyses

The GraphPad Prism 9 software (version 9.5.0) and Microsoft Excel for Microsoft 365 MSO (version 2211) were used to analyze the data. All numerical values were presented as mean ± standard error of the mean (SEM). Differences among more than three groups were analyzed using one-way analysis of variance followed by Tukey's Kramer tests. Statistical differences between two groups were analyzed using the Student's one-tailed $t$-test. Significant levels were set at $*P < 0.05$, $**P < 0.01$, $***P < 0.001$, and $****P < 0.0001$.

## Reporting summary

Further information on research design is available in the Nature Portfolio Reporting Summary linked to this article.

## Data availability

All data supporting the findings of this study are available within the paper and supplementary information files. Source data are provided with this paper.

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

## Acknowledgements

We thank A. Abe for their care of the laboratory animals. We appreciate the access to the slide scanner VS200 (Olympus) granted by the Research Core of Tokyo Medical and Dental University. This research was supported by the Basic Science and Platform Technology Programs for Innovative Biological Medicine (18am0301003h0005) and Advanced Biological Medicine (23am0401006h0005) to T.Y., from the Japan Agency for Medical Research and Development (AMED) and a JSPS KAKENHI Grant-in-Aid for Scientific Research (A) (19H01016 to T.N. and T.Y.), (A) (22H00440 to T.N.) and (B) (16H05221 to T.N.) from the Ministry of Education, Culture, Sports, Science and Technology (MEXT) of Japan (Tokyo). This research was also supported by the Joint Research Fund with Takeda Pharmaceutical Company, Limited.

## Author contributions

T.N., J.H., and T.Y. designed the research. J.H., T.N., K.I., J.T., S.E., K.Y.-T., M.Y., M.O., T.I., R.I.-H., M.N., K.M., A.S., M.N., S.Y., and F.S. performed the experiments and analyzed data. T.N., and J.H., wrote the manuscript. T.N. and T.Y. coordinated and supervised the project. All authors have read and approved the final manuscript.

## Competing interests

T.Y. has ongoing collaborations with Takeda Pharmaceutical Co., Ltd. and serves as an academic advisor for Rena Therapeutics Inc. The other authors declare no competing interests. M.N. and S.S. are paid employees of Takeda Pharmaceutical Company Limited.

## Additional information

[1]Department of Neurology and Neurological Science, Graduate School of Medical and Dental Sciences, Tokyo Medical and Dental University, 1-5-45 Yushima, Bunkyo-ku, Tokyo, 113-8519 Tokyo, Japan. [2]Center for Brain Integration Research, Tokyo Medical and Dental University, 1-5-45 Yushima, Bunkyo-ku, Tokyo, 113-8519 Tokyo, Japan. [3]NucleoTIDE and PepTIDE Drug Discovery Center, Tokyo Medical and Dental University, 1-5-45 Yushima, Bunkyo-ku, Tokyo, 113-8519 Tokyo, Japan. [4]Department of Bio-informational Pharmacology, Medical Research Institute, Tokyo Medical and Dental University, 1-5-45 Yushima, Bunkyo-ku, Tokyo, 113-8519 Tokyo, Japan. [5]Department of Cardiovascular Medicine, Tokyo Medical and Dental University, 1-5-45 Yushima, Bunkyo-ku, Tokyo, 113-8519 Tokyo, Japan. [6]Department of Cell Physiology, The Jikei University School of Medicine, 3-25-8, Nishi-Shimbashi, 105-8461 Minato-ku, Tokyo, Japan. [7]COE for Drug Metabolism, Pharmacokinetics and Modeling, Preclinical and Translational Sciences, Research, Takeda Pharmaceutical Company Limited, 2-26-1, Fujisawa, Kanagawa 251-8555, Japan. [8]Department of Materials Engineering, Graduate School of Engineering, University of Tokyo, 7-3-1 Hongo, Bunkyo-ku 113-8656 Tokyo, Japan. [9]These authors contributed equally: Juri Hasegawa, Tetsuya Nagata. ✉e-mail: t-naga.nuro@tmd.ac.jp; tak-yokota.nuro@tmd.ac.jp

