## [Peer Review File · Nature Communications]

REVIEWER COMMENTS

Reviewer #1 (Remarks to the Author):

In recent years the PMO chemistry has emerged as the leader for exon skipping applications in the muscle, but delivery is a major challenge due to poor PK properties that require massive doses for even modest activity. Several groups are addressing this using peptide- or antibody-conjugated PMO with very encouraging results, although no direct comparison to the present data is possible. In this study the authors demonstrate a novel approach to greatly enhance the PK and in vivo activity of PMO oligos by using the HDO design that they have previously demonstrated in several other models using other chemistries for the antisense strand. The duration and increased efficacy with high uniformity of expression across the muscle fibers are quite impressive, as is the safety, so the results are of considerable interest. The experiments appear to be well performed and the results generally support the conclusions.

Points of interest:

- Clever approach to try the HDO design with PMO! I wouldn't have thought this would work so well
- Demonstration of increased protein binding and improved in vivo PK – low protein binding is perhaps the single greatest deficiency of PMO, so this is an important finding supporting the rest of the work
- The use of diverse approaches to measure functional restoration in multiple tissues enhances the significance
- SC and IV dosing are feasible
- Demonstration of the requirement for cleavage of the complementary strand by either RNase or DNase for improved skipping activity of the HDO

Questions:

1. In Figure 1, why is the Toc-HDO only included in panels A, B, E? why not also in C, D? I assume the PK studies
2. The dose response curves in extended Figure 2 are quite nice, but the lowest dose of the Chol-HDO (25 mg/kg) already has a very high level of skipping, so I wonder how the curve looks at lower doses

Minor points:

1. The term “truncated” is often used incorrectly in referring to the protein or RNA that are generated by exon skipping – the term refers to something that has been shortened at an end, but in exon skipping this occurs internally, and so a more correct term would be “internally deleted”
2. The phrase “overwhelming skipping effect” in referring to the comparison between PMO and Chol-HDO is rather subjective – I’d prefer a more objective description of the relative difference.
3. The discussion refers to “improved prognosis for patients with DMD, which could not be achieved by PMO drugs” I don’t believe this is supportable by any data, so I suggest deleting this. Obviously the PMO have only a modest effect in humans, but the clinicians most active in prescribing PMO for patients with DMD are adamant that it improves their prognosis! And of course there are 4 approved PMO drugs, so the statement is not consistent with this....
4. “Improving the freezing behavior could lead to the treatment of psychological symptoms in patients with DMD” in the discussion should be rephrased to something like “Improving the freezing behavior could be associated with improving CNS / psychological symptoms in patients with DMD” although it is somewhat of a conjecture that these may be related mechanistically (as opposed to a secondary effect of treating the disease outside the CNS).
5. Please reword this incomplete sentence: DNase in case of use natural DNA instead of natural RNA in center portion of the complementary strand.”
6. Please replace “tolerance” with “resistance” (or something similar) in this phrase: “high tolerance of 2'-OMe RNA to nucleases”
7. Figure 3 panel B has a typo in “quadriceps”

Reviewer #2 (Remarks to the Author):

Hasegawa et al. present a study using heteroduplex oligonucleotide (HDO) technology as a treatment for DMD in the mdx mouse model. This group has previously published on the use of HDO technology. The advance in this manuscript is that the HDO entity consists of a PMO oligonucleotide as the therapeutic payload, with a 2'OMe/DNA mixed PS/PO backbone passenger oligonucleotide conjugated to a lipophilic delivery-assisting moiety. The PMO is a steric block oligo that promotes exon skipping and dystrophin translation reading frame restoration.

Overall, this manuscript reports a lot of data, with key experiments performed in terms of PK, dystrophin re-expression, histopathology, muscle/cardiac/CNS physiology, and toxicology. However, the manuscript has a number of deficiencies, which are outlined below.

Specific Criticisms/Comments

1. The data presented in this manuscript suggests that PMO/HDO-Chol oligos are vastly superior to molar equivalents of PMO alone. The data clearly support this. However, an obvious question is whether the improvements are due to the HDO format, HDO-Chol conjugates, or the presence of the cholesterol (or in some cases tocopherol) moiety. It seems that obvious controls are missing. Most notably, direct conjugation of PMOs with cholesterol should be tested at a minimum. I note that similar controls were tested in the previous manuscript (Nishina et al. Nat Comms 2015) where direct conjugation with tocopherol ablated gapmer ASO activity. However, steric block ASOs are quite distinct. Indeed, cholesterol conjugated steric block ASOs (antagomiRs) have been used to inhibit miRNAs for almost 2 decades.

2. The manuscript refers to the dystrophin-associated protein (DAP). However, this is confusing and I believe incorrect. I believe the authors are referring to the Dystrophin-associated protein complex (DAPC), also known as the dystro-glycan complex (DGC).

3. The authors state:

'Since the restored levels of the truncated dystrophin protein in skeletal muscles is approximately 6% of the normal dystrophin level'

This is an overstatement. After extensive analysis, dystrophin restoration following eteplirsen treatment was shown to be ~0.9% of healthy levels. Golodirsen is similar. Casimersen is a little higher, as exon 45 exhibits some degree of natural exon skipping. For viltolarsen, slightly higher numbers have been reported. A more balanced view of the state of exon skipping therapies is warranted, as this provides the central motivation for developing enhanced ASO technologies.

4. The authors state:

'Delivery and efficacy to the heart and diaphragm muscles, which have been difficult to achieve in preclinical studies, are particularly problematic'

This statement is not true/not complete. There has been extensive development of delivery assisted PMOs which can achieve high levels of dystrophin exon skipping in the heart. Specifically, peptide-PMO conjugates, and more recently antibody-oligonucleotide conjugates. There have been ~10 years of preclinical studies in this area, and recent clinical programs sponsored by Sarepta, PepGen, Entrada, Dyne, and Avidity.

It is true that naked PMO exhibits very low heart skipping activity, even at high doses. The authors should be more specific and not ignore large parts of the literature.

5. The authors state:

'Patients with DMD show symptoms of developmental, cognitive, learning, and behavioral difficulties owing to the defect of dystrophin expressed in CNS'

This is an overstatement. This is true in some patients. Not all DMD patients exhibit CNS-associated pathology.

6. The authors cite a Sarepta press release in reference to PPMO development. Why not cite the actual literature related to this technology?

7. The authors write the following:

'The HDO comprises a gapmer type ASO duplexed with a complementary RNA strand conjugated to a lipid ligand for delivery'

This is confusingly written in this context, as 'the' implies they are referring to the HDOs in the present manuscript. However, here I believe they are referring to their previous work. In short, containing a gapmer ASO is not an essential component of an HDO. Perhaps should be written as:

'Previously, we have reported a HDO technology whereby ...'

8. Gene nomenclature is used incorrectly. Gene and RNA symbols should be italics

a. 'PMOs targeting *Dmd* exon 23'

9. The authors write:

'however, was digested by RNase A (Extended Data Fig. 1A)'

Which I believe should be:

'however, [it] was digested by RNase A (Extended Data Fig. 1A)'

In reference to the data in Ex Fig 1A. I found this figure quite confusing. RNase A treatment looks undigested to me. Has this gel image been presented upside down? Size markers, or labels could clarify this issue. It is also not clear what kind of gel has been run here. I presume agarose, and fluorescence signal visualised?

10. In Fig 1B is the marker base pairs or nucleotides?

11. In several places, the authors refer to DLS data which is not included in the manuscript and so impossible to assess. This is important because the authors also state in the discussion:

'Particle formation was suggested to be involved in the enhanced uptake of tcDNA 17 and peptide PMO 51 into the tissues; however, the DLS results showed that no particle formation was observed in PMO/HDO.'

Nanoparticle formation is highly dependent on the biological matrix in which the oligos reside. Were DLS measurements made in PBS or serum, for example? The statement from the authors is impossible to judge due to a lack of data and methodological details.

12. In Fig 1C/D it is not clear what the comparisons are to. Is PMO naked PMO, or PMO complexed with cRNA-chol? Is Chol-HDO the PMO-HDO? There is a disparity between the diagrams in Fig 1A (which are quite clear) and the way the oligos are referred to in the text (which is confusing). The text in the main manuscript is confusing because it refers back to previous work on the gapmer-HDO technology. But it reads as if the gapmer-HDO was used as a control for the PK experiment.

13. The authors state:

'Furthermore, the blood profile of RNase A-untreated PMO/HDO indicates the relatively slow dissociation of PMO from PMO/HDO in the serum (Extended Data Fig. 1B).'

What are these data 'relative' to? No control data are provided, so these claims are difficult to assess.

14. Findings on ISH PMO distribution in treated animals are very interesting.

15. Exon skipping is assessed by semi-quantitative RT-PCR. While this method provides a useful qualitative visualisation of skipped transcript products, the field has largely moved away from this technique. A better alternative is RT-qPCR with absolute quantification (or digital PCR) to better quantify skipping. The reason for this suggestion, is that for the semi-quantitative RT-PCR there are multiple amplicons in the same reaction. These compete for reaction components and may exhibit different PCR efficiencies. Shorter amplicons, likely experience a reaction efficiency advantage, meaning that this technique will tend to over-estimate skipping levels.

16. Figures are presented in an unusual manner. These should have been assembled into a single montage for each figure.

17. The authors write:

'The decrease in the CNF was mild 29, however, it may improve with long-term treatment (Fig. 3D)'

This is a well-described phenomenon, whereby nuclei remain centrally-nucleated even after successful muscle regeneration in mouse muscle. As such CNF% is a good measure of 'historical regeneration'. I would not expect a reduction in CNF% after exon skipping over the time-frames used in this study. The statement that the CNF% might 'improve with long-term treatment' seems to be reaching, and no data is presented. It would be better for the authors to just describe their data, rather than trying to coerce the data to fit a narrative.

18. The authors write:

'Production of dystrophin was confirmed using western blot analysis (Fig. 3F). We found that PMO/HDOs restored markedly higher levels of dystrophin than PMOs in the heart and QF, where levels reached approximately 50% and 100%, respectively, compared with those in wild-type B10 control mice.'

This is a very bold statement to make. This is my biggest problem with this manuscript. Based on the blot images, I do not agree with the claimed levels of dystrophin restoration. The vinculin signal suggests that loading in the standards is less than in the samples, and so the dystrophin quantification has been overstated. (The standard curves should be mixtures of WT and mdx mice in defined ratios). No actual quantification is provided. It seems the authors just eyeballed their results as 100%, which is completely unacceptable and inconsistent with community standards in this field.

19. The authors should be very careful to not to over-interpret cardiac pathology findings in the mdx mice at this relatively young age. It is generally considered that the cardiac phenotype is very mild in this model.

20. No quantification of cardiac fibrosis is provided.

21. The authors state:

‘Approximately 5% exon-skipping was detected in the brain (Fig. 5A), which was similar to previous findings 17.’

This is a confusing statement. I believe the authors mean ‘similar to the level of exon skipping observed by Goyenvalle et al using tricycloDNA.’

22. The authors state:

‘Significant improvements in the grip test and treadmill test performance and in serum CK levels were also observed (Fig. 6B–D)’

This is not an accurate statement. In Fig 6C there is not a significant difference between Chol-HDO sc and mdx. The statistical test for B10 vs Chol-HDO iv is not shown. Authors need to be more specific about their statements.

23. The authors state:

‘intracellular cleavage of the complementary strand must be required to produce exon skipping in the heart and skeletal muscles.’

I don't agree with this. An alternative possibility is that the 2'OMe oligos bind more tightly to the PMO and block its activity. No mechanistic evidence is shown to support the claim that the passenger oligo cleavage occurs, or that this is important for activity. Notably, the data here are also not entirely consistent. The OMe gap HDO significantly improved Grip strength in Fig 7D despite negligible limb muscle skipping. This finding is a little puzzling to explain.

24. The authors have investigated potential renal toxicity of their HDO oligos. However, these are largely insufficient. Serum renal biomarkers are presented. ALT and AST are also markers of liver and muscle damage. They are not providing any useful information regarding renal toxicity. Kidney histology and urinary biomarkers such as Kim-1 would be more appropriate assessments of renal damage.

25. Data on CAV3 expression are presented and not discussed anywhere in the manuscript.

26. I was excited to see comparisons between the HDO technology and several other technologies. This would be the single most important comparison for this technology. The mdx mouse is perhaps one of the most 'cured' animal models in medical science. So studies that can restore dystrophin in mdx tissue are not interesting per se. However, demonstrating that there are important improvements between technologies is of high interest. I was therefore disappointed to realise that the authors are comparing their results to published findings, rather than performing head-to-head comparisons. Given the questions regarding the methods used to assess exon skipping and dystrophin re-expression raised above, I don't find the comparisons made by the authors in the discussion section to carry much weight.

Minor Language Issues

27. These mechanisms involve [the] increase in binding to serum albumin cleavage of the complementary strand to activate [the] PMO

28. affinity for serum albumin by lipid-conjugation [lipid-conjugated] ASO

Reviewer #3 (Remarks to the Author):

The manuscript by Hasegawa et al (#416645) describes a novel technology for in vivo delivery of heteroduplex oligonucleotides to improve the efficacy of induced RNA splice switching (exon skipping), compared to the use of PMOs (phosphorodiamidate morpholino oligomers). The manipulation of RNA-splice site selection has been considered a therapeutic strategy to skip an exon carrying a non-sense mutation and restore reading frame for protein production (albeit an internally truncated form). This strategy is of particular interest and focus in the case of Duchenne Muscular Dystrophies (DMD), which are caused by heterogeneous mutations of the Dystrophin gene, typically by deletions that lead to joining of out of frame exons and C-terminally truncated Dystrophin proteins. Splice site switching to restore reading frame and produce internally truncated and functional (to a certain degree) Dystrophin proteins has been considered a way to ameliorate the DMD pathology. The mdx mouse (carrying a nonsense mutation in exon 23 of the mouse Dystrophin gene) has been a primary model for testing various pre-translational strategies. One such strategy is the use of PMOs to block splicing to the mutated exon thus inducing splicing to following exon(s) to restore reading frame. While PMOs have been approved by the FDA for safety, they suffer from low efficiency and short duration in vivo.

In this work, the authors designed and tested several oligonucleotide compositions and modifications to boost serum retention, tissue uptake, and splice site switching efficacy. These innovations include RNA-RNA hybrid, RNA-DNA hybrid, RNA-2'OME hybrid, and with different 5' nucleotide and backbone chemical modifications (phosphorodiamidate and phosphorothioate). Different delivery and assessment time points were evaluated for serum and tissue retention, splice site switching efficiency, serum creatine kinase levels, immuno-histological improvement, and functional improvement (grip strength and treadmill exhaustion time). Multiple skeletal muscle groups as well as a few other selected organs were assessed by relevant assays. Dystrophin production was confirmed by immunostaining in vivo and western blotting. Most importantly, they also evaluated improvement of brain (by behavior) and cardiac functions.

Overall, they provide strong and convincing data that intravenous injection of the cholesterol (CHO)-modified heteroduplex oligonucleotides (PMO hybridized to complimentary CHO-DNA or CHO-RNA) show the best promise in the mdx model. I strongly support its publication as the information are important to those engaged in DMD research and clinical translation. Of note, DMD patients have different and complex mutations than a simple point mutation of the mdx model. Personalized designs of splice site switching oligo sequences will be more complicated. Nevertheless, the proof of principle data presented here is exciting.

There are two comments that I hope the authors will address to improve the presentation.

1. The statistical analyses were only done by t-test between 'specified' paired groups. This is insufficient/inappropriate when multiple groups are presented together. At minimal, ANOVA followed by Tukey's Kramer tests - given that some groups for comparison have different sample sizes). Even with their t-tests, there are missing comparisons between select pairs.

2. Animal numbers used are largely satisfactory. However, in a few graphs, the animal number appears low (i.e. $n=3$, judging by the dots in the graphs). While I understand that a considerable number of mice were already used throughout this study, boosting animal numbers to above 4 per assay/group will be more convincing and assuring as a pre-translational study in a high-profile journal.

Response to the Reviewer' comments

Reviewer #1:

In recent years the PMO chemistry has emerged as the leader for exon skipping applications in the muscle, but delivery is a major challenge due to poor PK properties that require massive doses for even modest activity. Several groups are addressing this using peptide- or antibody-conjugated PMO with very encouraging results, although no direct comparison to the present data is possible. In this study the authors demonstrate a novel approach to greatly enhance the PK and in vivo activity of PMO oligos by using the HDO design that they have previously demonstrated in several other models using other chemistries for the antisense strand. The duration and increased efficacy with high uniformity of expression across the muscle fibers are quite impressive, as is the safety, so the results are of considerable interest. The experiments appear to be well performed and the results generally support the conclusions.

Comment 1: 1. In Figure 1, why is the Toc-HDO only included in panels A, B, E? why not also in C, D? I assume the PK studies

Response:

Thank you for your suggestion.

We measured the concentration of PMO in the blood and muscles of mice administered Toc-HDO (11.88 $\mu\text{mol/kg}$). Additionally, to enhance the robustness of our findings, we have increased the sample size from 2 to N=4. The new results have been included as New Figures 1C and 1D.

New Figure 1C

New Figure 1D

Furthermore, based on these new results, we have made modifications to the main text as follows:

From

To investigate the pharmacokinetics (PK) of PMO/HDO in the serum of *mdx* mice, we administered systemic intravenous (IV) injections of 100 mg/kg PMO doses or the molar equivalent of cholesterol-conjugated cRNA (11.88 $\mu\text{mol/kg}$) with PMOs (Chol-HDO).

To

To investigate the pharmacokinetics (PK) of PMO/HDO in the serum of *mdx* mice, we administered systemic intravenous (IV) injections of **single** 100 mg/kg PMO doses or the molar equivalent of **tocopherol-conjugated cRNA (11.88 $\mu\text{mol/kg}$) hybridized with PMOs (Toc-HDO) or cholesterol-conjugated cRNA (11.88 $\mu\text{mol/kg}$) hybridized with PMOs (Chol-HDO)**. (Page 8, Line 11-14)

From

Chol-HDO exhibited a 4.8-fold greater drug exposure than PMOs ($\text{AUC}_{0-24\text{h}}$, 22389 nM/h vs. 4619 nM/h).

To

Compared to PMOs ($\text{AUC}_{0-24\text{h}}$, 3012.4 nM/h), Chol-HDO ($\text{AUC}_{0-24\text{h}}$, 21217.8 nM/h) exhibited a 7-fold greater drug exposure, while Toc-HDO showed a 5-fold greater exposure ($\text{AUC}_{0-24\text{h}}$, 15488.0 nM/h). (Page 9, Line 6-8)

Comment 2: *The dose response curves in extended Figure 2 are quite nice, but the lowest dose of the Chol-HDO (25 mg/kg) already has a very high level of skipping, so I wonder how the curve looks at lower doses.*

Response: We wish to express our appreciation to the reviewer's insightful comments. We have evaluated the skipping efficiency when we injected PMO at 10 mg/kg or the molar equivalent of Chol-HDO. These results have been included to the dose-response curve in the New extended Figure 2A.

(A)

New Extended Figure 2A

Minor points 1: *The term “truncated” is often used incorrectly in referring to the protein or RNA that are generated by exon skipping – the term refers to something that has been shortened at an end, but in exon skipping this occurs internally, and so a more correct term would be “internally deleted”*

Response:

Thank you for your suggestion.

The term " truncated dystrophin " in the manuscript has been replaced with " internal deleted dystrophin ". (Page 4, Line 11) (Page 5, Line 10) (Page 5, Line 15-6)

Minor points 2: *The phrase “overwhelming skipping effect” in referring to the comparison between PMO and Chol-HDO is rather subjective – I’d prefer a more objective description of the relative difference.*

Response:

Thank you for your suggestion.

We have changed the description.

From

Chol-HDO showed an overwhelming skipping effect compared with PMO in all muscle tissues investigated.

To

On maximum, Chol-HDO treatment induced 6.1-7.5-fold higher levels of skipping in skeletal muscles and the diaphragm and 22.7-fold higher levels in the heart than PMO. (Page 11 Line 7-9)

Minor points 3: *The discussion refers to “improved prognosis for patients with DMD, which could not be achieved by PMO drugs” I don’t believe this is supportable by any data, so I suggest deleting this. Obviously the PMO have only a modest effect in humans, but the clinicians most active in prescribing PMO for patients with DMD are adamant that it improves their prognosis! And of course there are 4 approved PMO drugs, so the statement is not consistent with this....*

Response:

Thank you for your suggestion.

We have changed the description.

From

Moreover, the normalization of cardiac dysfunction with robust expression of dystrophin in the heart of mdx mice suggests an improved prognosis for patients with DMD, **which could not be achieved by PMO drugs.**

To

Moreover, the normalization of cardiac dysfunction with robust expression of dystrophin in the heart of mdx mice suggests an improved prognosis for patients with DMD (Page 19, Line 18).

Minor points 4: *“Improving the freezing behavior could lead to the treatment of psychological symptoms in patients with DMD” in the discussion should be rephrased to something like “Improving the freezing behavior could be associated with improving CNS / psychological symptoms in patients with DMD” although it is somewhat of a conjecture that these may be related mechanistically (as opposed to a secondary effect of treating the disease outside the CNS).*

Response:

Thank you for your suggestion.

We have rephrased the sentence in the discussion as follows:

From

Improving the freezing behavior could lead to the treatment of psychological symptoms in patients with DMD.

To

Improving the freezing behavior could **be associated with improving CNS/psychological symptoms** in patients with DMD. (Page 20, Line 6)

Minor points 5: *Please reword this incomplete sentence: DNase in case of use natural DNA instead of natural RNA in center portion of the complementary strand.*

Response:

We would like to thank the reviewer for pointing out our error.

We have corrected this incomplete sentence as follows:

From

DNase in case of use natural DNA instead of natural RNA in center portion of the complementary strand.

To

DNase would cleave the complementary strand in case of use natural DNA instead of natural RNA in center portion of the complementary strand. (Page 22, Line 9)

Minor points 6: *Please replace “tolerance” with “resistance” (or something similar) in this phrase: “high tolerance of 2'-OMe RNA to nucleases”*

Response:

Thank you for your suggestion.

We have replaced “tolerance” with “resistance” as follows:

Additionally, PMO/HDOs consisting of a complementary strand fully composed of 2'OMe modifications showed no *in vivo* skipping activity, likely because the complementary strand was not cleaved owing to the high **resistance** of 2'-OMe RNA to nucleases. (Page 22, Line 13)

Minor points 6: *Figure 3 panel B has a typo in “quadriceps”*

Response:

Thank you for your point out.

We have corrected in this figure as follows:

Quadriceps

Reviewer #2:

Hasegawa et al. present a study using heteroduplex oligonucleotide (HDO) technology as a treatment for DMD in the mdx mouse model. This group has previously published on the use of HDO technology. The advance in this manuscript is that the HDO entity consists of a PMO oligonucleotide as the therapeutic payload, with a 2'OMe/DNA mixed PS/PO backbone passenger oligonucleotide conjugated to a lipophilic delivery-assisting moiety. The PMO is a steric block oligo that promotes exon skipping and dystrophin translation reading frame restoration.

Overall, this manuscript reports a lot of data, with key experiments performed in terms of PK, dystrophin re-expression, histopathology, muscle/cardiac/CNS physiology, and toxicology. However, the manuscript has a number of deficiencies, which are outlined below.

Comment 1: *The data presented in this manuscript suggests that PMO/HDO-Chol oligos are vastly superior to molar equivalents of PMO alone. The data clearly support this. However, an obvious question is whether the improvements are due to the HDO format, HDO-Chol conjugates, or the presence of the cholesterol (or in some cases tocopherol) moiety. It seems that obvious controls are missing. Most notably, direct conjugation of PMOs with cholesterol should be tested at a minimum. I note that similar controls were tested in the previous manuscript (Nishina et al. Nat Comms 2015) where direct conjugation with tocopherol ablated gapmer ASO activity. However, steric block ASOs are quite distinct. Indeed, cholesterol conjugated steric block ASOs (antagomiRs) have been used to inhibit miRNAs for almost 2 decades.*

Response:

We would like to express our sincere gratitude for the Reviewer's insightful comments, which have significantly contributed to the improvement of our paper.

1) HDO without a lipid ligand

When administering only HDO without a lipid ligand at a molar equivalent of 100 mg/kg of PMO (11.88 $\mu\text{mol/kg}$), the results are as shown in the following figure (Figure L1). Similar to PMO alone, there is almost no skipping effect in the heart. In the diaphragm and skeletal muscle, the skipping effect was lower compared to PMO alone. When compared to cholesterol conjugated HDO, it is evident that HDO without a ligand has a significantly lower skipping effect in the tissues we investigated. These results of HDO without a lipid ligand have been included to the dose-response curve in the New Extended Figure 2A (red square).

Figure L1

New Extended Figure 2A

2) Direct conjugation of PMOs with cholesterol

Thank you also for your remarks on cholesterol direct conjugated PMO.

We have been interested in this as well. There have been several reports on the binding of peptides to PMOs^{1, 2, 3, 4}, but there have been no reports on lipid ligands, as far as we have been able to find.

This required a great deal of time in searching for the necessary compounds, importing them to Japan, reaction conditions, isolation, etc. **We will discuss the details below, but in summary, although the synthesis of cholesterol directly conjugated to PMO was successful, the cholesterol conjugated PMO could not be administered to *mdx* mice due to its insolubility in water.**

We first decided to purchase the following two compounds: a mouse exon 23-targeting PMO conjugated with cyclooctyne (molecular weight 9004.4, see Figure L2) and Cholesteryl-TEG azide (molecular weight 630.9, see Figure L3).

Figure L2

Figure L3

In the case of the click reaction coupling of peptides and PMO, the peptide was dissolved in DMSO. However, Cholesteryl-TEG azide did not dissolve at all in DMSO, DMF or acetonitrile, so both Cholesteryl-TEG azide and PMO were finally dissolved in 100% ethanol.

Cyclooctyne-PMO and Cholesteryl-TEG azide were reacted in a molar ratio of 1:20 overnight with stirring. Then, we analyzed Cyclooctyne-PMO and its reaction solution with Cholesteryl-TEG azide using MALDI-TOF Mass Spectrometry (MS). The mass-ion peak around 9630 suggested that the Cholesteryl-TEG-PMO was synthesized.

After concentration, the reaction solution is dissolved in a 50% acetonitrile solution and analysed by reversed-phase high-performance liquid chromatography (RP-HPLC), because the reaction mixture could not be dissolved in water. At the same time, a Cyclooctyne-PMO aqueous solution is also analysed. The RP-HPLC chart for the Cyclooctyne-PMO is shown in Fig. L4A and the reaction solution in Fig. L4B.

(A) Cyclooctyne-PMO in H₂O

(B) Reaction solution in 50% acetonitrile

Figure L4

In Figure L4A, a peak at 15 minutes for Cyclooctyne-PMO is observed. Meanwhile, in Figure L4B, in addition to the peak of Cyclooctyne-PMO, a new peak at 23 minutes can be seen. This new peak has a maximum absorption wavelength of 260 nm, which is similar to that of Cyclooctyne-PMO, suggesting it is derived from PMO, and is thought to be Cholesteryl-TEG-PMO.

Reaction solution in H₂O

Figure L5

Next, although the reaction mixture was not completely dissolved in water as mentioned above, the supernatant was analyzed by HPLC in the same analytic conditions. As shown in Figure L5, the peak of Cyclooctyne-PMO was confirmed, but the peak at 23 minutes was not observed, indicating that Cholesteryl-TEG-PMO does not dissolve in aqueous solution.

As you pointed out, we have been using ASOs directly conjugated with lipid ligands for comparative studies with HDOs. These ASOs utilize phosphorothioate and phosphodiester linkages in their internucleotide bonds, which are negatively charged. This negative charge is thought to confer high water solubility, enabling these ASOs to dissolve effectively even when lipid ligands are conjugated. In contrast, Morpholinos (PMOs) use phosphorodiamidate linkages for their internucleotide bonds,

which are neutral. Consequently, PMOs are likely to have lower water solubility, potentially leading to insolubility when lipid ligands are directly conjugated. **This experience highlighted the benefit of attaching lipids to PMO with a complementary strand, as in heteroduplex oligonucleotides.**

We have added the following sentence in the result section.

We also administered unliganded PMO/HDO at a dose of 100 mg/kg (11.88 μ mol/kg). Similar to PMO, in the heart, there was almost no skipping activity, and in skeletal muscle and diaphragm, the skipping activity was slightly lower compared to PMO (Extended Data Fig. 2A red square). PMO conjugated directly with cholesterol could be synthesized. However, due to their high lipophilicity, they did not dissolve in water, thereby preventing their administration to mdx mice. This experience highlighted the benefit of attaching lipids to PMO with a complementary strand, as in heteroduplex oligonucleotides. (Page 12, Line 11-18)

Comment 2: *The manuscript refers to the dystrophin-associated protein (DAP). However, this is confusing and I believe incorrect. I believe the authors are referring to the Dystrophin-associated protein complex (DAPC), also known as the dystro-glycan complex (DGC).*

Response:

Thank you for your important suggestion.

The term "the dystrophin-associated protein (DAP)" in the manuscript has been replaced with "Dystrophin-associated protein **complex (DAPC)**".

(Page 5, Line 6) (Page 5, Line 8) (Page 14, Line 2) (Page 14, Line 4) (Page 19, Line 13)

Comment 3: *The authors state:*

'Since the restored levels of the truncated dystrophin protein in skeletal muscles is approximately 6% of the normal dystrophin level'

This is an overstatement. After extensive analysis, dystrophin restoration following eteplirsen treatment was shown to be ~0.9% of healthy levels. Golodirsen is similar. Casimersen is a little higher, as exon 45 exhibits some degree of natural exon skipping. For viltolarsen, slightly higher numbers have been reported. A more balanced view of the state of exon skipping therapies is warranted, as this provides the central motivation for developing enhanced ASO technologies.

Response:

Thank you for your important suggestion.

We have corrected as follows:

From

Since the restored levels of the truncated dystrophin protein in skeletal muscles is approximately 6% of the normal dystrophin level

To

Since the restored levels of the truncated dystrophin protein in skeletal muscles is 0.9-6 % of the normal dystrophin level (Page 5, Line 16)

Comment 4: *The authors state:*

'Delivery and efficacy to the heart and diaphragm muscles, which have been difficult to achieve in preclinical studies, are particularly problematic'

This statement is not true/not complete. There has been extensive development of delivery assisted PMOs which can achieve high levels of dystrophin exon skipping in the heart. Specifically, peptide-PMO conjugates, and more recently antibody-oligonucleotide conjugates. There have been ~10 years of preclinical studies in this area, and recent clinical programs sponsored by Sarepta, PepGen, Entrada, Dyne, and Avidity.

It is true that naked PMO exhibits very low heart skipping activity, even at high doses. The authors should be more specific and not ignore large parts of the literature.

Response:

Thank you for your important suggestion.

This section only refers to the naked PMO, so we have corrected as follows:

From

Delivery and efficacy to the heart and diaphragm muscles, which have been difficult to achieve in preclinical studies, are particularly problematic.

To

Delivery and efficacy to the heart and diaphragm muscles, which have been difficult to achieve by **naked PMO** in preclinical studies, are particularly problematic. (Page 5, Line 19)

Additionally, while the next paragraph discusses a series of peptide and antibody conjugated PMOs, we have added 2 new references 24 and 25, there.

reference 24

Klein AF, Varela MA, Arandel L, Holland A, Naouar N, Arzumanov A, Seoane D, Revillod L, Bassez G, Ferry A, Jauvin D, Gourdon G, Puymirat J, Gait MJ, Furling D, Wood MJ. Peptide-conjugated oligonucleotides evoke long-lasting myotonic dystrophy correction in patient-derived cells and mice. *J Clin Invest.* 2019:4739-4744. doi: 10.1172/JCI128205.

reference 25

Li X, Kheirabadi M, Dougherty PG, Kamer KJ, Shen X, Estrella NL, Peddigari S, Pathak A, Blake SL, Sizensky E, Genio CD, Gaur AB, Dhanabal M, Girgenrath M, Sethuraman N, Qian Z. The endosomal escape vehicle platform enhances delivery of oligonucleotides in preclinical models of neuromuscular disorders. *Mol Ther Nucleic Acids.* 2023 33:273-285. doi:10.1016/j.omtn.2023.06.022.

Comment 5: *The authors state:*

'Patients with DMD show symptoms of developmental, cognitive, learning, and behavioral difficulties owing to the defect of dystrophin expressed in CNS'

This is an overstatement. This is true in some patients. Not all DMD patients exhibit CNS-associated pathology.

Response:

Thank you for your important suggestion.

We have corrected as follows:

From

Patients with DMD show symptoms of developmental, cognitive, learning, and behavioral difficulties owing to the defect of dystrophin expressed in CNS ¹,

To

Some patients with DMD show symptoms of developmental, cognitive, learning, and behavioral difficulties owing to the defect of dystrophin expressed in CNS ¹, (Page 6, Line 1)

Comment 6: *The authors cite a Sarepta press release in reference to PPMO development. Why not cite the actual literature related to this technology?*

Response:

Thank you for your important suggestion.

We have removed a Sarepta press release and cite the following paper as reference 23. (Page 6, Line 10-11)

reference 23 Gan L, Wu LCL, Wood JA, Yao M, Treleaven CM, Estrella NL, Wentworth BM, Hanson GJ, Passini MA. A cell-penetrating peptide enhances delivery and efficacy of phosphorodiamidate morpholino oligomers in mdx mice. *Mol Ther Nucleic Acids*. 2022 Aug 17;30:17-27. doi: 10.1016/j.omtn.2022.08.019. PMID: 36189424; PMCID: PMC9483789.

Comment 7: *The authors write the following:*

'The HDO comprises a gapmer type ASO duplexed with a complementary RNA strand conjugated to a lipid ligand for delivery'

This is confusingly written in this context, as 'the' implies they are referring to the HDOs in the present manuscript. However, here I believe they are referring to their previous work. In short, containing a gapmer ASO is not an essential component of an HDO. Perhaps should be written as:

'Previously, we have reported a HDO technology whereby ...'

Response:

From

The HDO comprises a gapmer-type ASO duplexed with a complementary RNA strand conjugated to a lipid ligand for delivery²⁴.

To

Previously, we have reported a HDO technology⁵ whereby HDO comprises a ASO duplexed with a complementary RNA strand conjugated to a lipid ligand for delivery. (Page 5, Line 13)

Comment 8: *Gene nomenclature is used incorrectly. Gene and RNA symbols should be italics a. 'PMOs targeting Dmd exon 23'*

Response:

Thank you for your important suggestion.

We have corrected as follows:

We initially designed PMO/HDO comprising PMOs targeting *Dmd* exon 23 and its complementary strand conjugated with alpha-tocopherol or cholesterol ligand on the 5'-terminal end (Fig. 1A), and

confirmed the double-strand formation of both strands (Fig. 1B and Extended Data Table 1). (Page 8, Line 2)

Comment 9: *The authors write:*

'however, was digested by RNase A (Extended Data Fig. 1A)'

Which I believe should be:

'however, [it] was digested by RNase A (Extended Data Fig. 1A)'

In reference to the data in Ex Fig 1A. I found this figure quite confusing. RNase A treatment looks undigested to me. Has this gel image been presented upside down? Size markers, or labels could clarify this issue. It is also not clear what kind of gel has been run here. I presume agarose, and fluorescence signal visualised?

Response:

Thank you for your point out.

We have corrected as follows:

From

however, was digested by RNase A (Extended Data Fig. 1A)'

To

however, **it** was digested by RNase A (Extended Data Fig. 1A)' (Page 8, Line 6)

We apologize for the unclear explanation.

We performed electrophoresis using a 20% acrylamide gel. The 5' end of PMO is conjugated with Alexa647 (as shown in the figure L6),

Figure L6

and we have detected it through fluorescence. We have also run a 20 bp DNA marker at the same time, but it was not visible because it is not fluorescently labeled. After detecting Alexa647 labeled PMO or Chol-HDO, we confirm the 20 bp DNA marker by staining with GelRed. PMO did not migrate in electrophoresis due to its neutral charge as shown in Fig. 1B (lane 2). However, when Alexa-647 was conjugated to PMO, it became migratory due to the 4 negative charges (Figure L7) of Alexa-647 (Extended Data Fig. 1A Lane 1). Hybridization a complementary strand to Alexa-647-PMO resulted

in increased migration distance compared to Alexa-647-PMO alone due to the 24 negative charges of the complementary strand's PO⁻ and PS⁻ (Lane 2). When Chol-HDO (PMO) was treated with RNase A, the complementary strand was cleaved and dissociates. Therefore, the decrease of the negative charge from Alexa-647-Chol-HDO (Alexa-647-PMO alone) resulted in a shorter electrophoretic migration distance (Lane 4).

Figure L7

Comment 10: *In Fig 1B is the marker base pairs or nucleotides?*

Response:

Thank you for your point out.

The markers in Fig 1B represent base pairs.

Comment 11: *In several places, the authors refer to DLS data which is not included in the manuscript and so impossible to assess. This is important because the authors also state in the discussion:*

'Particle formation was suggested to be involved in the enhanced uptake of tcDNA 17 and peptide PMO 51 into the tissues; however, the DLS results showed that no particle formation was observed in PMO/HDO.'

Nanoparticle formation is highly dependent on the biological matrix in which the oligos reside. Were DLS measurements made in PBS or serum, for example? The statement from the authors is impossible to judge due to a lack of data and methodological details.

Response:

Thank you for your point out.

Here, we were measuring PMO and HDOs in PBS using a Zetasizer Pro (Malvern Panalytical Ltd). The histogram of the results has been included as New Extended Figure 2B. The histograms for PMO and HDOs show a narrow size distribution with diameters of less than 10 nm, suggesting that no aggregates have been formed. We have revised the text as follows:

From

PMO/HDO was readily dissolved in aqueous solutions and did not aggregate or form nanoparticles similar to PMOs, as confirmed via dynamic light scattering (DLS).

To

PMO/HDO was readily dissolved in aqueous solutions. To quantify particle sizes, dynamic light scattering measurements of PMO and HDOs in PBS were conducted. The histograms for PMO and HDOs show a narrow size distribution with diameters of less than 10 nm (Extended Figure 2B), suggesting that no aggregates have been formed (Page 8 line 7-11).

New Extended Data Figure 1B

Comment 12: *In Fig 1C/D it is not clear what the comparisons are to. Is PMO naked PMO, or PMO complexed with cRNA-chol? Is Chol-HDO the PMO-HDO? There is a disparity between the diagrams in Fig 1A (which are quite clear) and the way the oligos are referred to in the text (which is confusing). The text in the main manuscript is confusing because it refers back to previous work on the gapmer-HDO technology. But it reads as if the gapmer-HDO was used as a control for the PK experiment.*

Response:

We apologize for any confusion caused by these explanations."

1) In this paper, the main strand of the HDOs used is all PMOs, as shown in Figure 1A. The bolded part of the following sentence has caused confusion.

“Chol-PMO/HDO (RNase A treated: Total PMO) showed increased plasma retention (slower absorbance) and slower clearance compared with PMOs **similar to the gapmer-type HDO (Fig. 1C)** 27.”

In Figure 1C, the blood profile of Gapmer-HDO is not included; instead, we are comparing to the previously published blood profile data of Gapmer-HDO (Reference 27) as shown below.

Extended data figure 3d from Reference 27

(blood profile data of Cholesterol conjugated Gapmer-HDO)

We have revised the text as follows:

From

Chol-PMO/HDO (RNase A treated: Total PMO) showed increased plasma retention (slower absorbance) and slower clearance compared with PMOs similar to the gapmer-type HDO (Fig. 1C)²⁷.

To

the total PMO concentration of the RNase A treated serum from *mdx* mice administered Chol-PMO/HDO and Toc-PMO/HDO (RNase A treated: Total PMO) showed increased plasma retention (slower absorbance) and slower clearance compared with PMOs (Fig. 1C), similar to the gapmer-type HDO (Fig. 1C)²⁷. (Page 9 line 2-6)

2) Next, we will also provide an explanation of the PMO measurements.

We consider that in the case of HDO, as shown in Figure L8, there are three forms present in tissues and serum: a) naked PMO alone (without complementary strands), b) Chol-HDO (PMO) (Complete form), and c) Chol-HDO (PMO) with partially nuclease-cleaved complementary strand.

Figure L8

In the HELISA method we are using here, since same sequence as the complementary strand is used as a probe, we cannot measure HDOs with complementary strands. **Therefore, we can only measure the naked PMO a) and not the b) and c) forms.**

In Figure 1D, we measured only naked PMO in tissues administered with PMO, Toc-HDO (main strand is PMO), and Chol-HDO (main strand is PMO). In tissues where naked PMO is administered, all PMOs can be measured, but in tissues administered with Toc-HDO (PMO) and Chol-HDO (PMO), we can only measure naked PMO, so we cannot measure b) and c) forms. We consider that this naked PMO in the tissue directly contributes to exon skipping. We have changed the y-axis label to 'Naked PMO concentration' and also updated the legend notation.

New Figure 1D

In the previous Extended Figure 1B (New Extended Figure 1C), same as Figure 1D, we measured only naked PMO in serum administered with Toc-HDO (PMO) and Chol-HDO (PMO). We have changed the y-axis label to 'Naked PMO concentration' and also updated the legend notation.

previous Extended Figure 1B (New Extended Figure 1C)

Next, we wanted to evaluate all forms a), b) and c) in serum administered with Toc-HDO (PMO) and Chol-HDO (PMO). For this, we treated the HDOs with RNase A same way as shown in Extended Figure 1A. After this treatment, all three forms a), b) and c) became naked PMO a) form. Then, we measured this naked PMO, the results of which are in New Figure 1C. It represents the total amount of PMO (The sum of PMO from a), b), and c) forms) in the serum. We have changed the label on the y-axis to "Total PMO concentration." and also updated the legend notation.

New Figure 1C

We also have revised the text as follows:

From

Since double-stranded PMO/HDO cannot be measured using this method due to the complementary strand, the collected samples were pretreated with RNase A.

To

In this HELISA method, double-stranded PMO/HDO cannot be measured due to the presence of the complementary strand. Therefore, to measure all PMOs in the serum from *mdx* mice administered Toc-HDO or Chol-HDO, the collected samples were pretreated with RNase A to completely remove the complementary strand and convert them into naked PMOs before measuring them using the HELISA method. (Page 8, Line 16-Page 9 Line 1)

Comment 13: *The authors state:*

'Furthermore, the blood profile of RNase A-untreated PMO/HDO indicates the relatively slow dissociation of PMO from PMO/HDO in the serum (Extended Data Fig. 1B).'

What are these data 'relative' to? No control data are provided, so these claims are difficult to assess.13.

Response:

Thank you for pointing that out.

We have completely misunderstood. Until now, as for ligand conjugated gapmer-type HDO, we have never observed the blood profile of a single-stranded ASO dissociating from ligand conjugated gapmer-type HDO in the blood (Reference 27), so we couldn't make a relative comparison.

We have revised the text as follows:

From

Furthermore, the blood profile of RNase A-untreated PMO/HDO indicates the relatively slow dissociation of PMO from PMO/HDO in the serum (Extended Data Fig. 1B)

To

The blood profile of RNase A-untreated Chol-PMO/HDO indicates **a slow** dissociation of PMO from PMO/HDO in the **blood** (see Extended Data Fig. 1B), **rather than an immediate dissociation following injection**. (Page 9 Line 9-11)

Comment 14: *Findings on ISH PMO distribution in treated animals are very interesting.*

Response:

Thank you for your interest.

Furthermore, we have also conducted in situ hybridization chain reaction (ISH HCR)^{5,6} to evaluate the nuclear localization of PMO (Cy5 probe) in the heart and quadriceps femoris (QF) 6 hours after a single injection of either PMO or Chol-HDO at a concentration of 11.88 $\mu\text{mol/kg}$. To assess the nuclear localization precisely, we simultaneously performed staining using antibodies against lamin A/C (labeled with Alexa 488) for the nuclear membrane and wheat germ agglutinin lectin (labeled with Alexa 594) for the cell membrane, along with DAPI counterstaining. The framed section in the left figure is magnified in the middle (showing quadruple staining with Cy5, Alexa 488, Alexa 594, and DAPI) and right figures (showing Cy5 and Alexa 488 staining). In the heart and quadriceps muscles treated with PMO, the PMO signal was scarcely detectable. In contrast, in the both muscles treated with Chol-HDO, there was a pronounced localization of PMO within both the cytoplasm and nuclei. These results have been included as the New Extended Figure 1F and 1G.

New Extended Figure 1F and 1G

Comment 15: *Exon skipping is assessed by semi-quantitative RT-PCR. While this method provides a useful qualitative visualisation of skipped transcript products, the field has largely moved away from this technique. A better alternative is RT-qPCR with absolute quantification (or digital PCR) to better quantify skipping. The reason for this suggestion, is that for the semi-quantitative RT-PCR there are multiple amplicons in the same reaction. These compete for reaction components and may exhibit different PCR efficiencies. Shorter amplicons, likely experience a reaction efficiency advantage, meaning that this technique will tend to over-estimate skipping levels.*

Response:

Thank you for your important suggestion.

Verheul et al. reported that digital droplet PCR (ddPCR) provides the highest accuracy for evaluating exon skipping⁷. Additionally, Hiller et al. also reported that ddPCR is the most accurate method, followed by analyzing samples with single-round PCR followed by Agilent bioanalyzer (capillary electrophoresis)⁸. Therefore, as you pointed out, we believe that using ddPCR for evaluation provides the highest accuracy.

In papers from Sarepta Therapeutics and Dyne Therapeutics etc^{9, 10, 11}, all of which were reported after 2022, they still evaluated exon skipping using capillary electrophoresis after single-round PCR. Here, we are currently using single-round PCR followed by analysis with the Agilent Bioanalyzer 2100, and while it may lack some accuracy, we believe it hasn't had a significant impact on the results.

Due to equipment constraints with ddPCR, we initially tried the quantitative real-time PCR system¹². However, we encountered several issues. The major problem occurred when using PBS (No template control). Despite PBS being the template, fluorescence was detected with the primer/probe set designed to detect the Skip band, and the CT values were around 30. It would not have been an issue if the CT values of the samples were below 30, but they were distributed between 28-32. Some samples had higher CT values than PBS (indicating lower expression), which has made interpretation difficult, and we are currently re-evaluating our method.

Comment 16: *Figures are presented in an unusual manner. These should have been assembled into a single montage for each figure.*

Response:

Thank you for your point out.

The figures were arranged in this way to make it easier for reviewers to view. Once the paper is accepted, each figure will be assembled into a single montage.

Comment 17: *The authors write:*

'The decrease in the CNF was mild 29, however, it may improve with long-term treatment (Fig. 3D)'

This is a well-described phenomenon, whereby nuclei remain centrally-nucleated even after successful muscle regeneration in mouse muscle. As such CNF% is a good measure of 'historical regeneration'. I would not expect a reduction in CNF% after exon skipping over the time-frames used in this study. The statement that the CNF% might 'improve with long-term treatment' seems to be reaching, and no data is presented. It would be better for the authors to just describe their data, rather than trying to coerce the data to fit a narrative.

Response:

Thank you for your point out.

We have removed the phrase 'however, it may improve with long-term treatment' from the following text in the Results section.

The decrease in the CNF was mild ³², ~~however, it may improve with long-term treatment~~ (Fig. 3D). (Page 13 Line 19 -Page 14, Line 1).

Comment 18: *The authors write:*

'Production of dystrophin was confirmed using western blot analysis (Fig. 3F). We found that PMO/HDOs restored markedly higher levels of dystrophin than PMOs in the heart and QF, where levels reached approximately 50% and 100%, respectively, compared with those in wild-type B10 control mice.'

This is a very bold statement to make. This is my biggest problem with this manuscript. Based on the blot images, I do not agree with the claimed levels of dystrophin restoration. The vinculin signal suggests that loading in the standards in less than in the samples, and so the dystrophin quantification has been over-stated. (The standard curves should be mixtures of WT and mdx mice in defined ratios). No actual quantification is provided. It seems the authors just eyeballed their results as 100%, which is completely unacceptable and inconsistent with community standards in this field.

Response:

Thank you for your suggestion.

As per your suggestion, we have used a mixture of B10 and *mdx* samples in defined ratios (100:0, 50:50, 10:90) as a dystrophin control. As a result, the difference in expression between each control became more distinct compared to before (New Figure 3F). Also, we have analyzed the intensity of dystrophin and vinculin bands using Image J 1.54g (ROI manager) and graphed their ratios. Given that this Western Blot is based on chemiluminescence and lacks some linearity, and the controls only have one sample each, it is difficult to claim accuracy. Compared to B10, almost 100% expression of dystrophin was observed in the heart and skeletal muscles treated with Chol-HDO, while about 50% expression of dystrophin was observed in the heart and skeletal muscles treated with Toc-HDO.

New Figure 3F

Comment 19. *The authors should be very careful to not to over-interpret cardiac pathology findings in the mdx mice at this relatively young age. It is generally considered that the cardiac phenotype is very mild in this model.*

Response:

As you pointed out, in this *mdx* mouse model, unlike in patients with DMD, abnormalities are not observed in echocardiograms unless they are of a very old age (40 weeks or older). In this study, improvements in electrocardiographic abnormalities and myocardial fibrosis were observed, but we intend to interpret these findings cautiously. Therefore, we have revised the following two sentences in the discussion section.

From

In particular, Chol-HDO normalized the motor and cardiac phenotype, serum CK, and muscle pathology in *mdx* mice to their levels in B10 mice.

To

In particular, Chol-HDO normalized the motor and cardiac phenotype, serum CK, and muscle pathology in *mdx* mice to their levels in B10 mice, **although the cardiac phenotype of *mdx* mice is very mild.** (Page 19 Line 6)

From

Moreover, the normalization of cardiac dysfunction with robust expression of dystrophin in the heart of *mdx* mice ~~suggests an improved prognosis for patients with DMD.~~

To

Moreover, the normalization of cardiac dysfunction with robust expression of dystrophin in the heart of *mdx* mice **may indicate a potential for improved prognosis in patients with DMD.** (Page 19 Line 17-18)

Comment 20. *No quantification of cardiac fibrosis is provided.*

Response:

We apologies for the confusion.

The graph in Fig. 4F represents the quantitative analysis of cardiac fibrosis.

(F) Areas of heart fibrosis

Figure 4F

Comment 21. *The authors state:*

'Approximately 5% exon-skipping was detected in the brain (Fig. 5A), which was similar to previous findings¹⁷.'

This is a confusing statement. I believe the authors mean 'similar to the level of exon skipping observed by Goyenvalle et al using tricycloDNA.'

Response:

Thank you for your suggestion. We have changed the description.

From

Approximately 5% exon-skipping was detected in the brain (Fig. 5A), which was similar to previous findings¹⁷.

To

Similar to the level of exon skipping observed by Goyenvalle et al using tricycloDNA was detected in the brain (Fig. 5A)¹⁷. (Page 16, Line 4-5)

Comment 22. *The authors state:*

'Significant improvements in the grip test and treadmill test performance and in serum CK levels were also observed (Fig. 6B–D)'

This is not an accurate statement. In Fig 6C there is not a significant difference between Chol-HDO sc and mdx. The statistical test for B10 vs Chol-HDO iv is not shown. Authors need to be more specific about their statements.

Response:

We would like to thank the reviewer for pointing out.

We have corrected the following sentence as follows:

From

Significant improvements in the grip test and treadmill test performance and in serum CK levels were also observed (Fig. 6B–D)

To

Significant improvements were also observed in the grip test and in serum CK levels, but not in treadmill test performance (Fig. 6B–D). (Page 16, Line 15-17)

Statistical results comparing *mdx* intravenously administered with Chol-HDO and B10 (B10 vs Chol-HDO iv) have been included in Figures 6B, C and D (New Fig. 6B-D). Across all figures, no significant differences were observed between these two groups.

New Fig. 6B-D

Comment 23. *The authors state:*

'intracellular cleavage of the complementary strand must be required to produce exon skipping in the heart and skeletal muscles.'

I don't agree with this. An alternative possibility is that the 2'OMe oligos bind more tightly to the PMO and block its activity. No mechanistic evidence is shown to support the claim that the passenger oligo cleavage occurs, or that this is important for activity. Notably, the data here are also not entirely consistent. The OMe gap HDO significantly improved Grip strength in Fig 7D despite negligible limb muscle skipping. This finding is a little puzzling to explain.

Response:

Thank you for your point out.

The complementary strand with full 2'-OMe modifications exhibits significantly stronger binding to PMO, as shown in the T_m data of the Extended Data table 1. As you suggested, the absence of skipping activity may not be due to the complementary strand remaining uncleaved, but rather because the complementary strand not dissociating from the PMO or dissociating very slowly. It's possible that the dissociation varies by tissue, which could explain the unaccountable results seen in Fig 7D. Therefore, we have revised our expression as follows.

From

Since full 2'-OMe modifications of the complementary strand were stable more than 3 days post-intravenous injection in hepatocytes *in vivo*⁵, intracellular cleavage of the complementary strand must be required to produce exon skipping in the heart and skeletal muscles.

To

Since the complementary strand with full 2'-OMe modifications was stable for more than 3 days post-intravenous injection in hepatocytes *in vivo*⁵, intracellular cleavage **or dissociation** of the complementary strand must be required to produce exon skipping in the heart and skeletal muscles.

(Page 4 Line 15)

Comment 24. *The authors have investigated potential renal toxicity of their HDO oligos. However, these are largely insufficient. Serum renal biomarkers are presented. ALT and AST are also markers of liver and muscle damage. They are not providing any useful information regarding renal toxicity. Kidney histology and urinary biomarkers such as Kim-1 would be more appropriate assessments of renal damage.*

Response:

Thank you for your point out.

Unfortunately, we were unable to adequately collect mouse urine, so we have investigated the pathology of the kidneys and liver. In the Chol-HDO administered group, no obvious abnormalities were observed, however, in the Toc-HDO administered group, we observed an increase in size heterogeneity of hepatocytes in the liver and a slight increase of cellular density in glomeruli of the kidneys.

These results have been included as the New Extended Figure 5E.

We also have added the following sentence in the result section.

Additionally, we have conducted histological evaluations using hematoxylin and eosin staining of *mdx* mice treated with PBS, PMO, Toc-HDO, or Chol-HDO (Extended Fig. 5E). In the liver pathology (upper panel), small foci of inflammatory cell infiltration were observed in the liver parenchyma of *mdx* mice treated with any of the interventions, including PBS. No other significant lesions were observed in *mdx* mice treated with Chol-HDO. However, in *mdx* mice treated with Toc-HDO, there was occasional increased size heterogeneity of hepatocyte nuclei, sometimes accompanied by meganucleation. In kidney pathology (lower panel), no significant lesions were noted in *mdx* mice following treatment with PBS or Chol-HDO. On the other hand, *mdx* mice treated with Toc-HDO occasionally showed a slight increase in cellular density in glomeruli. (Page 18 Line 2-11)

Comment 25. *Data on CAV3 expression are presented and not discussed anywhere in the manuscript.*

Response:

We apologize for the lack of explanation.

The images of double staining with Caveolin3 and DAPI shown in Fig. 3E are representative images captured for assessing the cross-sectional area of myofibers and the centrally nucleated fibers in the QF. The explanation of these images was only provided in the legend and not elsewhere in the main text. We have revised the text as follows.

From

To evaluate the effect of robust dystrophin expression on histology, the myofiber cross-sectional area (CSA) and centrally nucleated fibers (CNF) of QF were measured as a marker of muscle regeneration

To

To evaluate the effect of robust dystrophin expression on histology, the QF was double stained with Caveolin3 and DAPI (Fig. 3E), and then the myofiber cross-sectional area and centrally nucleated fibers (CNF) of QF were measured as a marker of muscle regeneration^{32,33}. (Page 13 line 11-13)

Comment 26. *I was excited to see comparisons between the HDO technology and several other technologies. This would be the single most important comparison for this technology. The mdx mouse is perhaps one of the most 'cured' animal models in medical science. So studies that can restore dystrophin in mdx tissue are not interesting per se. However, demonstrating that there are important improvements between technologies is of high interest. I was therefore disappointed to realise that the authors are comparing their results to published findings, rather than performing head-to-head comparisons. Given the questions regarding the methods used to assess exon skipping and dystrophin re-expression raised above, I don't find the comparisons made by the authors in the discussion section to carry much weight.*

Response:

As reviewer pointed out, both the dosage and the days until analysis after last administration are different, so we cannot simply compare. We have also considered synthesizing these novel nucleic acid structures but have abandoned the effort due to the high level of technical difficulty and lack of information. We were able to synthesize the amidites for tcDNA, but unfortunately, the synthesis of the oligomer has not been going well.

Ultimately, we have not been able to head-to-head compare any of them. We have removed the relevant section from the discussion.

This time, to compare with these nucleic acids, we have administered Chol-HDO under the same experimental conditions.

1) First, to compare with Palmitate-tcDNA-PS¹², we have administered Chol-HDO at 10 µmol/kg/week for four times, and then sacrifice two weeks after the final administration. As shown below, we observed 46% skipping in the QF and 32% in the heart. It has been reported that 25% skipping in the QF and 10% in the heart (Figure 1D from Relizani et al)¹² was observed by administered Palmitate-tcDNA-PS, so the Chol-HDO are showing better results.

10 μ mol/kg/week for four times by Chol-HDO

2) Dyne Therapeutics administered FORCE-M23D at a dose equivalent to 30mg/kg PMO to mice and analyzed them one week later ¹⁰. When we administered and analyzed Chol-HDO under the same conditions, we observed 15% skipping in the heart and 18% in the QF. Mice treated with FORCE-M23D exhibited 40% skipping in the heart and 45% in the QF. The single dose results were better for FORCE-M23D than for Chol-HDO (Figure 3A and E from Desjardins et al) ¹⁰. Wave Life Science also administered their chimeric stereopure oligonucleotides with phosphorothioate and phosphoryl guanidinecontaining (PN) at 30 mg/kg and analyzed the gastrocnemius muscle one week later ¹³. The most effective, DMD-2788 (Figure 5B from Kandasamy et al.) ¹³, showed a 5% skipping efficiency in that muscle. Chol-HDO achieved 15% in the gastrocnemius muscle, but a direct comparison with DMD-2788 is not possible due to the unknown molecular weight of this compound.

Single injection by Chol-HDO at a dose equivalent to 30 mg/kg PMO

3) Sarepta Therapeutics administered Peptide-conjugated PMO (PPMO) at a dose equivalent to 40mg/kg PMO to mice and analyzed them one week later ⁹. This time, we also compared with PPMO. We administered Chol-HDO at a dose equivalent to 40mg/kg PMO. With PPMOs, 60% skipping in the QF and 13% in the heart were observed (Figure 3B from Gan et al) ⁹. When analyzed under the same conditions, Chol-HDO showed 18% skipping in the QF, but 15% in the cardiac muscle, indicating that Chol-HDO had slightly better results in the heart.

Single injection by Chol-HDO at a dose equivalent to 40 mg/kg PMO

As you have pointed out, we also believe that *mdx* mice are more amenable to treatment compared to human patients. However, in our study, even with high-dose administration, we achieved complete normalization of CK levels. Furthermore, most of the improvements in motor function reported in other studies were observed in grip strength tests. Sarepta Therapeutics has recently shown improvement in Rotarod tests, but in our study, we observed improvement up to the level of B10 in Treadmill tests. Treadmill tests, which require sustained running, necessitate enhancement of overall muscle and cardiac function. Considering this, we believe our results are highly significant.

Comment 27. Minor Language Issues

These mechanisms involve [the] increase in binding to serum albumin cleavage of the complementary strand to activate [the] PMO

Response:

Thank you for your point out.

We have revised the text as follows.

These mechanisms involve **the** increase in binding to serum albumin, thereby improving blood retention and cellular uptake with the release of lipid ligands inside cells via cleavage of the complementary strand to activate **the** PMO. (Page 4, Line13 and line 15)

Comment 28. Minor Language Issues

affinity for serum albumin by lipid-conjugation [lipid-conjugated] ASO

Response:

Thank you for your point out.

We have revised the text as follows.

Chappell et al. previously reported that increased affinity for serum albumin by ~~lipid-conjugation~~ **conjugated** ASO facilitates ASO transport across endothelial barriers into the interstitium of the muscle⁵⁰. (Page 21, Line 2)

Reviewer #3:

The manuscript by Hasegawa et al (#416645) describes a novel technology for in vivo delivery of heteroduplex oligonucleotides to improve the efficacy of induced RNA splice switching (exon skipping), compared to the use of PMOs (phosphorodiamidate morpholino oligomers). The manipulation of RNA-splice site selection has been considered a therapeutic strategy to skip an exon carrying a non-sense mutation and restore reading frame for protein production (albeit an internally truncated form). This strategy is of particular interest and focus in the case of Duchenne Muscular Dystrophies (DMD), which are caused by heterogeneous mutations of the Dystrophin gene, typically by deletions that lead to joining of out of frame exons and C-terminally truncated Dystrophin proteins. Splice site switching to restore reading frame and produce internally truncated and functional (to a certain degree) Dystrophin proteins has been considered a way to ameliorate the DMD pathology. The mdx mouse (carrying a nonsense mutation in exon 23 of the mouse Dystrophin gene) has been a primary model for testing various pre-translational strategies. One such strategy is the use of PMOs to block splicing to the mutated exon thus inducing splicing to following exon(s) to restore reading frame. While PMOs have been approved by the FDA for safety, they suffer from low efficiency and short duration in vivo.

In this work, the authors designed and tested several oligonucleotide compositions and modifications to boost serum retention, tissue uptake, and splice site switching efficacy. These innovations include RNA-RNA hybrid, RNA-DNA hybrid, RNA-2'OME hybrid, and with different 5' nucleotide and backbone chemical modifications (phosphorodiamidate and phosphorothioate). Different delivery and assessment time points were evaluated for serum and tissue retention, splice site switching efficiency, serum creatine kinase levels, immuno-histological improvement, and functional improvement (grip strength and treadmill exhaustion time). Multiple skeletal muscle groups as well as a few other selected organs were assessed by relevant assays. Dystrophin production was confirmed by immunostaining in vivo and western blotting. Most importantly, they also evaluated improvement of brain (by behavior) and cardiac functions.

Overall, they provide strong and convincing data that intravenous injection of the cholesterol (CHO)-modified heteroduplex oligonucleotides (PMO hybridized to complimentary CHO-DNA or CHO-RNA) show the best promise in the mdx model. I strongly support its publication as the information are important to those engaged in DMD research and clinical translation. Of note, DMD patients have different and complex mutations than a simple point mutation of the mdx model. Personalized designs of splice site switching oligo sequences will be more complicated. Nevertheless, the proof of principle data presented here is exciting.

Comment 1: *The statistical analyses were only done by t-test between 'specified' paired groups. This is insufficient/inappropriate when multiple groups are presented together. At minimal, ANOVA followed by Tukey's Kramer tests - given that some groups for comparison have different sample sizes). Even with their t-tests, there are missing comparisons between select pairs.*

Response:

We apologize for the insufficient description.

As for the analysis settings in GraphPad, in cases where there are different sample sizes for multiple group comparisons, ANOVA followed by Tukey's Kramer tests are automatically performed. This has been added to the Methods section.

Differences among more than three groups were analyzed using one-way analysis of variance followed by Tukey's Kramer tests.

(Page 34, Line 7)

Comment 2: *Animal numbers used are largely satisfactory. However, in a few graphs, the animal number appears low (i.e. n=3, judging by the dots in the graphs). While I understand that a considerable number of mice were already used throughout this study, boosting animal numbers to above 4 per assay/group will be more convincing and assuring as a pre-translational study in a high-profile journal.*

Response:

Thank you for your suggestion.

We have increased the number of mice to a minimum of four for all groups with fewer than three mice in Fig. 1C, Fig. 1D, Fig. 2B, Extended Data Fig. 1C and Extended Data Fig. 2A.

New Figure 1C

New Figure 1D

New extended Figure 1C

(B) exon 23-skipped dystrophin mRNA after 1, 3, or 5X weekly administration

New Figure 2B

New extended Figure 2A

Reference

1. Shabanpoor F, Gait MJ. Development of a general methodology for labelling peptide-morpholino oligonucleotide conjugates using alkyne-azide click chemistry. *Chem Commun (Camb)* **49**, 10260-10262 (2013).
2. Betts C, *et al.* Pip6-PMO, A New Generation of Peptide-oligonucleotide Conjugates With Improved Cardiac Exon Skipping Activity for DMD Treatment. *Molecular therapy Nucleic acids* **1**, e38 (2012).
3. Nikan M, *et al.* Targeted Delivery of Antisense Oligonucleotides Using Neurotensin Peptides. *J Med Chem* **63**, 8471-8484 (2020).
4. Wesolowski D, Tae HS, Gandotra N, Llopis P, Shen N, Altman S. Basic peptide-morpholino oligomer conjugate that is very effective in killing bacteria by gene-specific and nonspecific modes. *Proceedings of the National Academy of Sciences* **108**, 16582-16587 (2011).
5. Zhuang P, Zhang H, Welchko RM, Thompson RC, Xu S, Turner DL. Combined microRNA and mRNA detection in mammalian retinas by in situ hybridization chain reaction. *Sci Rep* **10**, 351 (2020).
6. Dirks RM, Pierce NA. Triggered amplification by hybridization chain reaction. *Proceedings of the National Academy of Sciences* **101**, 15275-15278 (2004).
7. Verheul RC, van Deutekom JCT, Datson NA. Digital Droplet PCR for the Absolute Quantification of Exon Skipping Induced by Antisense Oligonucleotides in (Pre-)Clinical Development for Duchenne Muscular Dystrophy. *PLoS One* **11**, e0162467 (2016).
8. Hiller M, *et al.* A multicenter comparison of quantification methods for antisense oligonucleotide-induced DMD exon 51 skipping in Duchenne muscular dystrophy cell cultures. *PLoS One* **13**, e0204485 (2018).
9. Gan L, *et al.* A cell-penetrating peptide enhances delivery and efficacy of phosphorodiamidate morpholino oligomers in mdx mice. *Molecular therapy Nucleic acids* **30**, 17-27 (2022).

10. Desjardins CA, *et al.* Enhanced exon skipping and prolonged dystrophin restoration achieved by TfR1-targeted delivery of antisense oligonucleotide using FORCE conjugation in mdx mice. *Nucleic Acids Res*, (2022).
11. Li X, *et al.* The endosomal escape vehicle platform enhances delivery of oligonucleotides in preclinical models of neuromuscular disorders. *Molecular therapy Nucleic acids* **33**, 273-285 (2023).
12. Relizani K, *et al.* Palmitic acid conjugation enhances potency of tricyclo-DNA splice switching oligonucleotides. *Nucleic Acids Res* **50**, 17-34 (2021).
13. Kandasamy P, *et al.* Control of backbone chemistry and chirality boost oligonucleotide splice switching activity. *Nucleic Acids Res* **50**, 5443-5466 (2022).

REVIEWERS' COMMENTS

Reviewer #1 (Remarks to the Author):

The revised manuscript has satisfactorily addressed my prior comments, and I have no new questions or comments that require further edits. The splice switching efficacy of the authors HDO PMO conjugates is noteworthy, and will be of broad interest for the field of nucleic acid therapeutics, given the strong safety profile.

Reviewer #2 (Remarks to the Author):

My comments have been very comprehensively addressed. My congratulations to the authors.

Reviewer #3 (Remarks to the Author):

The revised manuscript (416645) has addressed all my comments/critiques. I have reviewed the revised text and figures, and am satisfied with the additions and changes. The new data continue to support the original conclusion, and I highly recommend its publication.

Response to the Reviewer' comments

Reviewer #1 (Remarks to the Author):

The revised manuscript has satisfactorily addressed my prior comments, and I have no new questions or comments that require further edits. The splice switching efficacy of the authors HDO PMO conjugates is noteworthy, and will be of broad interest for the field of nucleic acid therapeutics, given the strong safety profile.

Response:

Thank you very much for dedicating your time and expertise to peer-review our manuscript. Your thorough analysis and constructive feedback have been invaluable in enhancing the quality and clarity of our work. We deeply appreciate your contributions to refining our study and helping us present our findings more effectively.

Reviewer #2 (Remarks to the Author):

My comments have been very comprehensively addressed. My congratulations to the authors.

Response:

We would like to express my sincere gratitude for the time and effort you have spent reviewing our manuscript. Your detailed and insightful feedback has significantly contributed to the improvement of our work. The comprehensive manner in which you addressed my comments not only helped in refining our paper but also in deepening our understanding of the subject matter. It is truly appreciable how your expertise has guided us towards presenting our research in the best possible light. Thank you for your valuable contribution to our work.

Reviewer #3 (Remarks to the Author):

The revised manuscript (416645) has addressed all my comments/critiques. I have reviewed the revised text and figures and am satisfied with the additions and changes. The new data continue to support the original conclusion, and I highly recommend its publication.

Response:

Thank you immensely for dedicating your expertise and time to review our manuscript. Your thorough examination and constructive feedback have been pivotal in enhancing the quality and impact of our study. The revisions made, guided by your insightful comments, have not only addressed the initial concerns but also enriched the manuscript significantly. We are particularly grateful for your endorsement of the work's publication.